# Cep55 promotes cytokinesis of neural progenitors but is dispensable for most mammalian cell divisions

Antonio Tedeschi [1,2 ✉], Jorge Almagro [1], Matthew J. Renshaw [3], Hendrik A. Messal [1,5], Axel Behrens [1,4] & Mark Petronczki [2,6]

In mammalian cell lines, the endosomal sorting complex required for transport (ESCRT)-III mediates abscission, the process that physically separates daughter cells and completes cell division. Cep55 protein is regarded as the master regulator of abscission, because it recruits ESCRT-III to the midbody (MB), the site of abscission. However, the importance of this mechanism in a mammalian organism has never been tested. Here we show that *Cep55* is dispensable for mouse embryonic development and adult tissue homeostasis. *Cep55*-knockout offspring show microcephaly and primary neural progenitors require Cep55 and ESCRT for survival and abscission. However, Cep55 is dispensable for cell division in embryonic or adult tissues. In vitro, division of primary fibroblasts occurs without Cep55 and ESCRT-III at the midbody and is not affected by ESCRT depletion. Our work defines Cep55 as an abscission regulator only in specific tissue contexts and necessitates the re-evaluation of an alternative ESCRT-independent cell division mechanism.

[1] Adult Stem Cell Laboratory, The Francis Crick Institute, 1 Midland Road, London NW1 1AT, UK. [2] Cell Division and Aneuploidy Laboratory, Clare Hall Laboratories, Cancer Research UK London Research Institute, London EN6 3LD, UK. [3] Advanced Light Microscopy, The Francis Crick Institute, 1 Midland Road, London NW1 1AT, UK. [4] Faculty of Life Sciences, King's College London, Guy's Campus, London SE1 1UL, UK. [5]Present address: Division of Molecular Pathology, Oncode Institute, Netherlands Cancer Institute, Plesmanlaan 121, 1066CX Amsterdam, The Netherlands. [6]Present address: Boehringer Ingelheim RCV GmbH & Co KG, A-1121 Vienna, Austria. ✉email: antonio.tedeschi@crick.ac.uk

To physically separate, daughter cells must resolve the thin intercellular bridge that connects them at the end of cytokinesis[1,2]. The endosomal sorting complex required for transport (ESCRT) mediates membrane remodeling in several cellular processes, including the formation of multivesicular bodies, nuclear envelope reformation, and cytokinetic abscission[3,4]. Signaling molecules and specific adaptors recruit the ESCRT complex at different sites of action in the cell. The early acting sub-complexes ESCRT-0, I, and II then recruit ESCRT-III components to form filamentous structures that promote membrane remodeling and scission. According to the current model of abscission, the adaptor protein Cep55 recruits the ESCRT-III sub-complex, which then promotes membrane constriction and scission of the intercellular bridge[2,5–10]. Inactivation of most ESCRT-III components (CHMP1A, B, CHMP2A, B, CHMP3, CHMP4A, B, C, CHMP5, 6, 7, and IST1) results in abscission defects in human cancer cell lines, including a severe delay in completion of cell division followed by furrow regression and the emergence of binucleated cells[8,11–15]. Despite extensive validation in cell lines, the relevance of ESCRT-mediated abscission is just beginning to be explored in multicellular organisms[5,16].

Inactivation of ESCRT components in *Drosophila*[17,18] and *Caenorhabditis elegans*[19] reveals that the complex is required in the specialized context of germ cell or embryonic divisions, but its importance in somatic divisions is unclear. In mice, defects in cell division have not been reported after deletion of ESCRT components[20–22]. However, due to the pleiotropic functions of ESCRT complexes and the multiplicity of complex components, the importance of the ESCRT-mediated abscission function in vivo might have been difficult to evaluate in the models studied to date[16].

The midbody (MB), a complex protein structure that contains bundles of microtubules at the center of the intercellular bridge, serves as a platform for the assembly of the abscission machinery[2]. The ESCRT-III sub-complex is recruited at both sides of the MB and extends in the form of helical structures to the abscission sites, where it is thought to pinch the plasma membrane[8–10,23–26]. In mammalian cell lines, the adaptor protein Cep55 recruits the ESCRT complex at the MB[6,7]. Cep55 association with the MB is tightly regulated by Plk1 phosphorylation at S436[27,28]. Only once Plk1 is degraded at the end of mitosis can the C terminus of Cep55 directly interact with the core MB protein MKLP1[28,29]. Cep55 can then recruit the ESCRT-associated protein ALIX (also known as PDCD6IP) and the ESCRT-I component Tsg101, which in turn assemble the ESCRT-III complex at the MB[6,7,12,13,30]. Cep55 depletion or mutation of key residues in its ESCRT- and ALIX-binding region (EABR) prevent ESCRT accumulation at the MB[6,7,30]. Cep55-depleted cultured human cells remain connected through an intercellular bridge for a protracted period and eventually become binucleated[6,7,27,29]. Therefore, inactivation of Cep55 in mammals would be expected to impair abscission more specifically than inactivation of ESCRT-III subunits.

Interestingly, biallelic Cep55-truncating mutations are responsible for two human syndromes named MARCH[31] and Meckel-like syndrome[32,33]. These pathologies exhibit similar clinical features including cerebral abnormalities and kidney defects in stillborn infants. However, these infants often retained the mutant Cep55 mRNA and in the case of MARCH patients the mutated Cep55 retained the EABR domain. Thus, how mammalian abscission is affected in vivo in the absence of Cep55 remains unknown.

Here we show using Cep55-knockout mice that Cep55 is dispensable for the divisions of many cell types in vivo. Unexpectedly, Cep55-knockout mice progress through embryogenesis and are born live. Despite Cep55's crucial role in the division of cancer cells in culture, most embryonic mouse tissues and the adult intestine are formed normally in Cep55-knockout animals. However, Cep55-knockout newborns show severe microcephaly, reduced brain cortical thickness, binucleated neurons, and kidney abnormalities, similar to the features described in human infants affected by MARCH and Meckel-like syndromes. These findings offer an explanation for the etiology of the human pathologies. Consistent with the in vivo results, Cep55- or Chmp4B-depleted neural progenitors fail abscission and become binucleated. In contrast, Cep55-null primary fibroblasts cultured in vitro do not recruit ALIX, Tsg101, Chmp2B, and Chmp4B at the MB, and can successfully divide when ESCRT-III components are depleted. Altogether, this work defines Cep55 as a specific abscission factor during brain development, revealing that a yet undefined Cep55- and ESCRT-III-independent mechanism mediates cell division in primary fibroblasts.

## Results

**Cep55-knockout mice are born live but die postnatally**. To investigate the relevance of Cep55-mediated abscission in vivo, we generated Cep55-null mice from embryonic stem (ES) cells obtained by the EUCOMM program[34]. The Cep55-null alleles used in this study were designed to prevent the expression of the C-terminal part of Cep55, including exon 6, which encodes key residues of the EABR, and exons 9–10, which encode the MB localization domain (Fig. 1a, b and Supplementary Fig. 1a, b). Unexpectedly, we found that intercrosses of mice harboring germline Cep55-null alleles (Cep55−) generated live Cep55−/− mice at approximately the expected Mendelian ratio, at embryonic stages (E)13.5–18.5 and at birth (P0) (Table 1). We observed that Cep55−/− newborn mice often lacked milk in their stomachs and were lighter than control littermates (average 82% of control body weight; Fig. 1c, d). However, Cep55−/− animals were also lighter than control littermates at E18.5 (average 82% of control body weight; Fig. 1d), indicating that the lack of milk in the stomach might not be the only explanation for differences in weight at P0. Postnatally, most Cep55−/− mice were cannibalized by their parents and the few survivors (9.5%, $n = 3$ out of 74) showed severe motor coordination problems (Supplementary Movie 1), which impaired feeding and therefore necessitated killing before weaning (Table 1).

To verify that Cep55 was deleted from these mice, we generated tail tip fibroblasts (TTFs) from newborn mice and mouse embryonic fibroblasts (MEFs) from E13.5 embryos. We could not detect any residual mRNA in TTFs using specific primers for exons 6–8, or exons 3–4, likely indicating that the mutant mRNA was degraded by nonsense-mediated decay (Supplementary Fig. 1c). Moreover, using an antibody against the C terminus of Cep55, we could not detect any residual protein in Cep55−/− TTFs, MEFs, or organs (Fig. 1e and Supplementary Fig. 1d–f). These results confirm that Cep55 is knocked out in these mice. We conclude that Cep55 deletion results in postnatal mortality, but that Cep55 is largely dispensable for mouse embryogenesis.

**Cep55-knockout mice show microcephaly and kidney dysmaturity**. Although the body morphogenesis of Cep55−/− newborn mice appeared normal, the skull appeared flat (Fig. 1c), suggesting possible microcephaly. Indeed, brains isolated from Cep55−/− E18.5 embryos and neonates were abnormally small, averaging ~40% of control brain weight (Fig. 2a, b). The mutant cortices appeared thinner and more transparent than control ones, and were more fragile during isolation from the skull. Histological sections showed ~50% decrease in the length, thickness, and cellularity of the Cep55−/− cortex during embryonic development and at birth, although the typical layered structure of the cortex

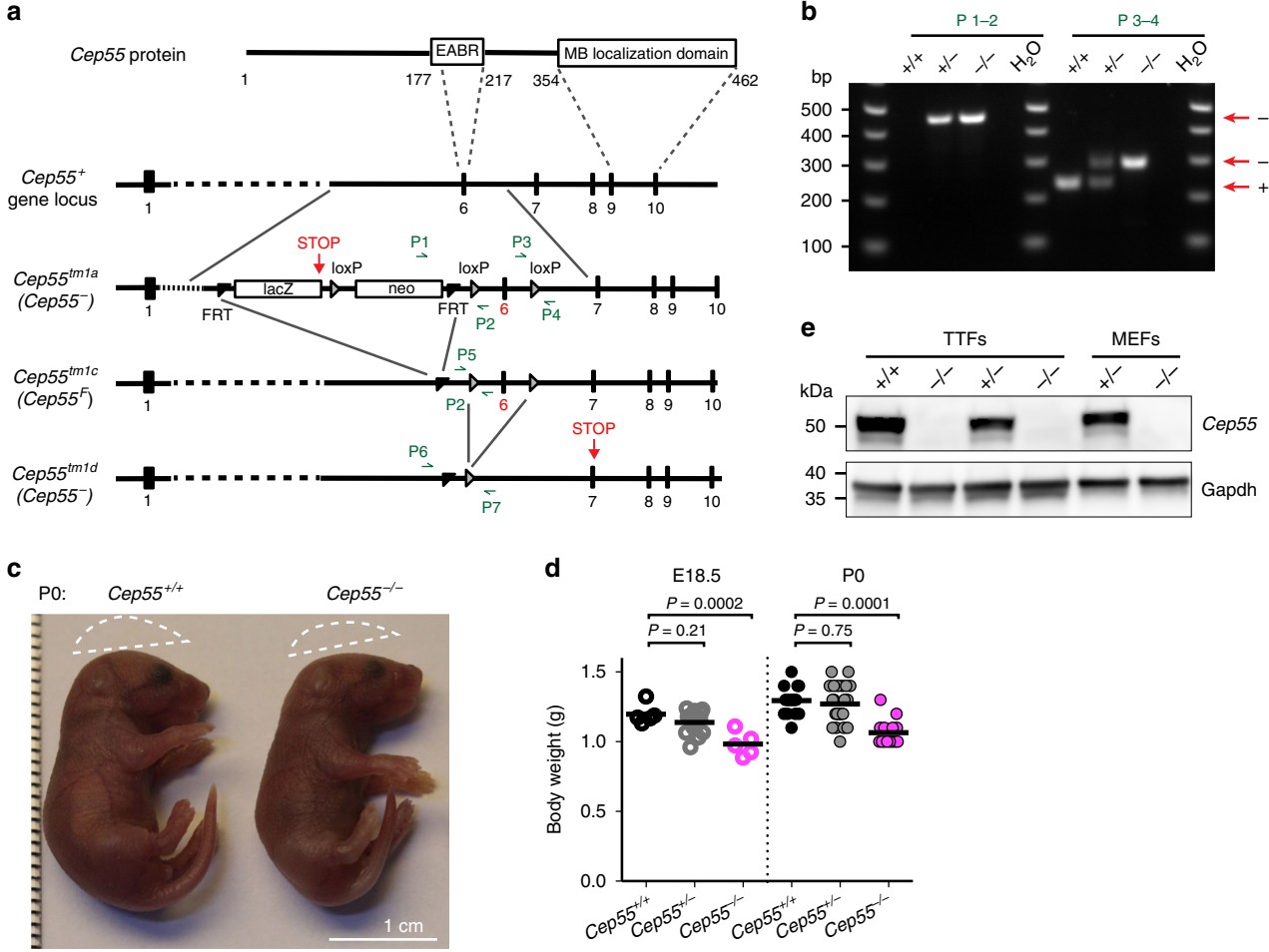

**Fig. 1 Cep55-knockout mice are viable. a** Schematic representation of mouse Cep55 protein domains (EABR, ESCRTs and ALIX-binding domain; MB, midbody) and *Cep55* genomic locus, showing wild-type allele (+), the knockout first allele tm1a (including the selection cassette (neo), the LacZ trapping cassette, and LoxP and FRT recombination sites), the conditional allele tm1c (F, floxed), and the deletion allele tm1d (−). **b** PCR analysis of primary mouse tail tip fibroblasts (TTFs) with primers P1–4 shown in **a** to verify *Cep55* status; *n* = 3 independent experiments. **c** Images of newborn (P0) mice of the indicated genotypes. Dotted lines indicate skull shape. **d** Body weights of mice of the indicated genotypes and developmental stages. Horizontal bars indicate mean; *n* = 5, 20, 5; 17, 24, and 14 mice, respectively; *P*-values calculated using one-way ANOVA followed by Dunnett's multiple comparisons test. **e** Western blott of protein extracts from TTFs and mouse embryonic fibroblasts (MEFs) of the indicated genotypes with antibodies against Cep55 and Gapdh; *n* = 3 independent experiments. *Cep55*−/− in **b**, **c**, **d**, and **e** indicates *Cep55*tm1a/tm1a mice. Source data for **e** and **d** are provided as a Source Data file.

**Table 1 Cep55-knockout mice are viable.**

| Genotype | Stage | Homozygous | Heterozygous | Wild-type | Total |
|---|---|---|---|---|---|
| tm1a/WT x tm1a/WT | P1–P14 | 7 (9.5%)* | 43 (58.1%) | 24 (32.4%) | 74 |
| | P0 | 28 (30.8%) | 46 (50.5%) | 17 (18.7%) | 91 |
| | E13.5–E18.5 | 23 (26.8%) | 48 (55.8%) | 15 (17.4%) | 86 |
| tm1b/WT** x tm1b/WT | P0 | 3 (20%) | 7 (46.7%) | 5 (33.3%) | 15 |
| | E13.5–E18.5 | 4 (12.9%) | 17 (54.8%) | 10 (32.3%) | 31 |
| tm1d/WT x tm1d/WT | E13.5–E18.5 | 24 (22.2%) | 55 (50.9%) | 29 (26.9%) | 108 |

*Cep55*-knockout mice are born at expected Mendelian ratios. Numbers (and percentages) of offspring of the indicated genotypes resulting from the crosses indicated on the left are given. Separate quantifications are given for offspring at different stages: P1–P14, postnatal day 1–14; P0, birth; E13.5–E18.5, embryonic day 13.5–18.5. *Homozygous *Cep55*tm1a/tm1a were culled at P1 (2 mice), P9 (2 mice), and P14 (3 mice). **See ref. [34] for the structure of the tm1b allele.

was maintained (Fig. 2c–h). Overall, these results are consistent with human data that describe cerebral cortex defects in infants carrying *Cep55*-truncating mutations[31].

In addition to brain defects, a range of kidney defects have been described in human infants with *Cep55* mutations[31–33]. When we analyzed the rare surviving P14 *Cep55*−/− mice (*n* = 3), we found that the tissue architecture of the mutant kidneys was different from that of age-matched control kidneys. Of note, the cortical thickness of *Cep55*−/− kidneys was reduced compared with controls and glomeruli appeared predominantly distributed in the superficial cortex (Supplementary Fig. 2a–c). This peripheral distribution is typically observed in the cortex of mice younger than P14[35]. Glomeruli in *Cep55*−/− mice were also smaller and hypercellular compared with those in control mice (Supplementary

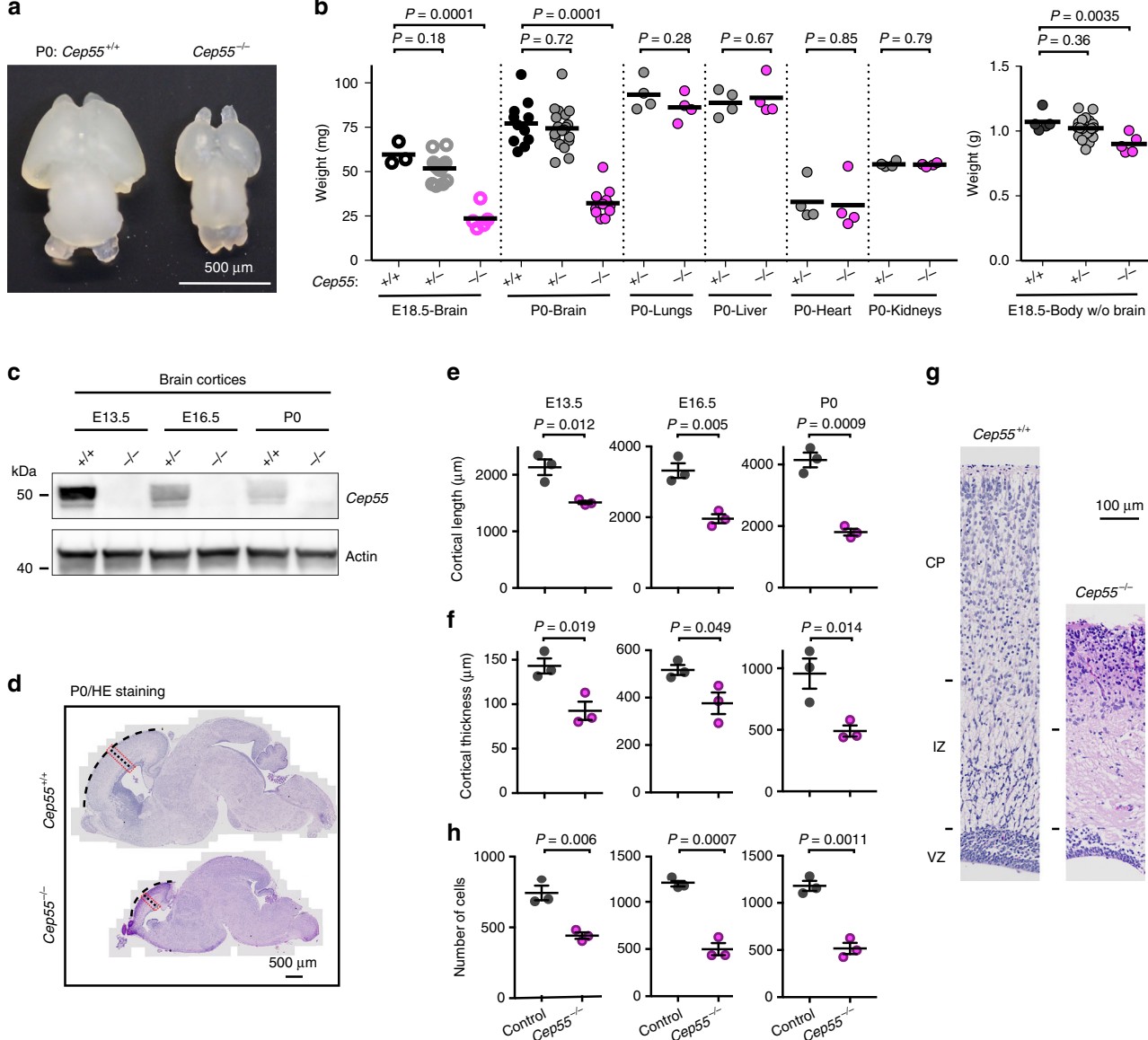

**Fig. 2 Cep55-knockout mice show microcephaly. a** Brain images from P0 mice of the indicated genotypes. **b** Weights of the indicated organs and genotypes. Horizontal bars indicate mean; $n = 3, 13, 5$ (E18.5-brain); 11, 20, 12 (P0-brain); 4 and 4 (P0-lungs, liver, heart, and kidneys); 5, 20, and 5 (E18.5-body without (w/o) brain) mice, respectively; $P$-values calculated using one-way ANOVA followed by Dunnett's multiple comparisons test where more than two sets of data are compared, otherwise Student's two-tailed unpaired $t$-test. **c** Western blot of protein extracts from brain cortices of the indicated genotypes and developmental stages with antibodies against Cep55 and Actin; $n = 3$ independent experiments. **d** Sagittal sections of P0 brains stained with hematoxylin and eosin (HE). Dotted black lines indicate cortical dimensions measured in **e** (curved) and **f** (straight). Dotted red boxes indicate area enlarged in **g. e, f** Quantification of cortical length (**e**) and cortical thickness (**f**) at the indicated developmental stages. Control includes Cep55$^{+/+}$ and Cep55$^{+/-}$ mice. $n = 3$ mice per genotype. **g** Enlarged view of the forebrain cortices from **d**. CP, cortical plate; IZ, intermediate zone; VZ, ventricular zone. **h** Cell counts in a 200 μm-wide field of neocortex in control and Cep55$^{-/-}$ mice. $n = 3$ mice per genotype. **e, f, h** Horizontal bars indicate mean; error bars indicate SEM; $P$-values calculated using Student's two-tailed unpaired $t$-test. **a–g** Cep55$^{-/-}$ indicates Cep55$^{tm1a/tm1a}$ mice (see Fig. 1a for allele details). Source data for **b, c, e, f, h** are provided as a Source Data file.

Fig. 2d, e). As the predominance of immature, fetal-type glomeruli and the reduced cortical thickness was unexpected for the age of these mice, we defined this condition as kidney dysmaturity.

**Loss of Cep55 promotes microcephaly as a result of apoptosis.**
To further investigate the cause of the defects observed in Cep55 mutant cortices, we first examined the expression of Cep55 in control animals. At E13.5, Cep55 was highly expressed in the mouse nervous system, including the cortex, shown by LacZ expression under the Cep55 endogenous promoter

(Supplementary Fig. 3a). Interestingly, western blotting showed that Cep55 protein level was higher at E13.5 than at later developmental stages (Fig. 2c and Supplementary Fig. 3b). Hematoxylin and eosin (HE) staining of brain sections revealed pyknotic nuclei, suggesting cell death, in Cep55$^{-/-}$ cortices at E13.5 and E16.5 (Fig. 3a–c). Indeed, staining for cleaved caspase 3 (active caspase 3, C3A) showed that up to 25% of cells were apoptotic in embryonic Cep55$^{-/-}$ cortices, in contrast to Cep55$^{+/+}$ controls that contained <0.2% apoptotic nuclei (Fig. 3d–h). By using the FLASH technique[36] to stain whole embryos, we confirmed that the cortex of mutant mice was highly

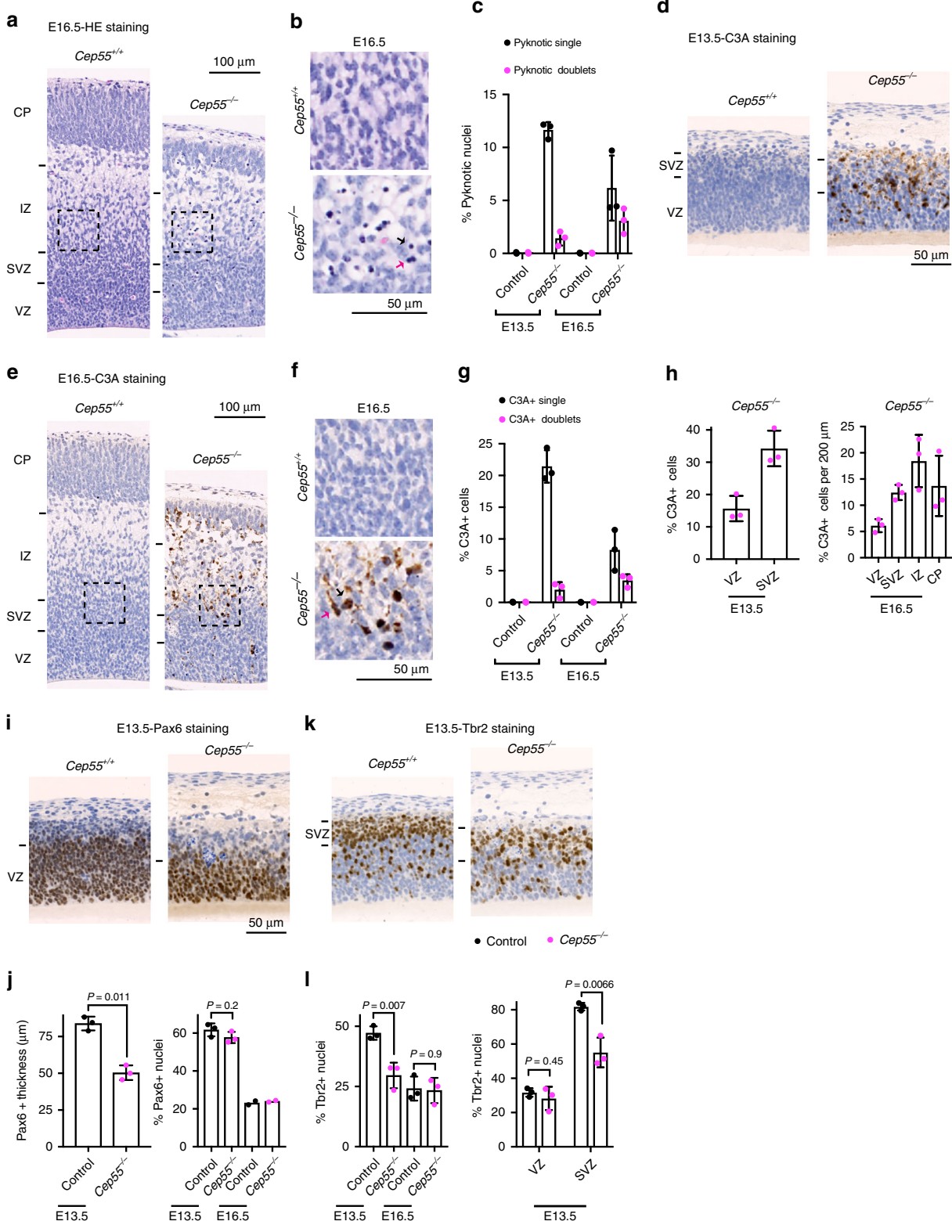

apoptotic (Supplementary Fig. 3c). The percentage of Ki67-positive nuclei scored by immunohistochemistry was not reduced in *Cep55*−/− cortices (Supplementary Fig. 3d, e), indicating that differences in proliferation do not explain the reduced cellularity in *Cep55*−/− cortices (Fig. 2h).

Next, we examined the neural progenitor population in *Cep55* mutant cortices by immunostaining for Pax6 and Tbr2 to mark apical and basal progenitor nuclei, respectively[37]. Despite a slight but significant increase in the percentage of proliferating cells at E13.5 (Supplementary Fig. 3d), the thickness of the Pax6-positive ventricular zone (VZ) was decreased in *Cep55*−/− cortices, suggesting that Cep55 might be required at earlier developmental stages (Fig. 3i, j). However, the percentage of Pax6-positive cells was similar between control and mutant cortices (Fig. 3i, j). The

**Fig. 3 *Cep55* deletion results in apoptotic cells and a reduction of neural progenitors in the brain cortex. a** HE staining of comparable regions of *Cep55*$^{+/+}$ and *Cep55*$^{-/-}$ cortices at E16.5. Boxed areas are magnified in **b**. Black arrow, single pyknotic nucleus; red arrow, pyknotic doublet. **c** Quantification of pyknotic nuclei in embryonic *Cep55*$^{+/+}$ and *Cep55*$^{-/-}$ cortices. n = 3 mice per genotype; 4039, 2330, 2189, and 1067 cells quantified, respectively. **d**, **e** Immunohistochemical staining of E13.5 (**d**) and E16.5 (**e**) cortices for active Caspase 3 (C3A). Boxed areas in **e** are magnified in **f**. Black arrow, single C3A-positive cell; red arrow, C3A-positive doublet. **g** Quantification of C3A-positive (C3A+) cells in embryonic *Cep55*$^{+/+}$ and *Cep55*$^{-/-}$ cortices. n = 3 mice per genotype; 4076, 2290, 2087, and 1123 cells quantified respectively. **h** Quantification of C3A+ cells in embryonic *Cep55*$^{-/-}$ cortical layers. n = 3 mice per genotype; 1123 and 2290 cells quantified, respectively. **i** Immunohistochemistry of *Cep55*$^{+/+}$ and *Cep55*$^{-/-}$ mouse cortices for the apical neural progenitor marker Pax6. **j** Quantification of the thickness of the Pax6-positive ventricular zone (Pax6+ thickness) (left) and the percentage of Pax6+ nuclei (right) in cortical sections like those in **i**. n = 3 mice per genotype at E13.5; 2 mice per genotype at E16.5. Cell numbers quantified for control and *Cep55*$^{-/-}$, respectively, were 2585 and 1238 (E13.5), and 2136 and 1238 (E16.5). **k** Immunohistochemistry of *Cep55*$^{+/+}$ and *Cep55*$^{-/-}$ mouse cortices for the basal neural progenitor marker Tbr2. **l** Left: quantification of the percentage of Tbr2-positive (Tbr2+) nuclei in cortical sections like those in **k**. n = 3 mice per genotype and per stage; cell numbers quantified for control and *Cep55*$^{-/-}$, respectively, were 1840 and 1080 (E13.5), and 3010 and 1835 (E16.5). Right: Distribution of Tbr2+ nuclei in the VZ and subventricular zone (SVZ). P-values calculated using Student's two-tailed unpaired t-test. All bar charts show mean ± SD. Source data for **c**, **g**, **h**, **j**, **l** are provided as a Source Data file.

percentage of Tbr2-positive basal progenitors, which are required for the expansion of the cerebral cortex[38], was reduced in Cep55 mutant cortices at E13.5 (Fig. 3k, l), in line with the reduced numbers of Pax6-positive progenitors. The sub-VZ (SVZ) was also less well defined, with fewer Tbr2-positive cells and aggregates of dead cells (Fig. 3k, l). Indeed, the SVZ showed a higher percentage of C3A-positive cells compared with the VZ at E13.5 (Fig. 3d, h). In addition to the SVZ, the intermediate zone (IZ) and the cortical plate (CP), which are occupied by neurons, were particularly affected by cell death at E16.5 (Fig. 3e, h). Together, these data are consistent with a role for Cep55 in the survival of neural progenitors, most notably Tbr2-positive progenitors, and neurons during embryonic neurogenesis.

**Binucleated cells are present in *Cep55*-knockout cortex.** Our observation that some pyknotic nuclei and C3A-positive cells appeared as doublets in *Cep55*$^{-/-}$ brain sections (Fig. 3b, c, f, g) is reminiscent of the binucleated cells resulting from *Cep55* knockdown in cancer cell lines[6,7,27,29]. To analyze binucleation of cells in the developing brain, we dissociated E16.5 cortices and analyzed cellular DNA content by propidium iodide staining and flow cytometry. The percentage of cells with a 4N DNA content increased from 2.56 ± 2.1% in control to 22.27 ± 0.53% in *Cep55*-deleted cortices and was confirmed with 4′,6-diamidino-2-phenylindole (DAPI) staining (Fig. 4a–c and Supplementary Fig. 4a). Deletion of *Cep55* also resulted in a significant increase in post-mitotic (Ki67-negative) cortical cells with a 4N DNA content (Fig. 4a–c; 0.7 ± 0.4% control vs. 12.1 ± 1.8% *Cep55*$^{-/-}$), consistent with binucleation. The percentage of total Ki67-positive cells was not different between control and *Cep55*-knockout samples (Fig. 4c), consistent with the immunohistochemistry data (E16.5; Supplementary Fig. 3d, e). Confocal imaging of mutant cortices revealed that 65% of pyknotic doublets (37 doublets, 6 mice) were composed of a binucleated cell pair connected by a cytoplasmic bridge (Fig. 4d–f). Many binucleated cells (44.4%; 18 cells, 3 mice) were also positive for C3A, suggesting that some cells die before cell division is completed (Fig. 4e, f). In newborn *Cep55*$^{-/-}$ mice, 32.2 ± 8.9% of neurons in the CP (mean ± SD; n = 284 cells from 3 mice) also appeared as doublets or were abnormally large compared with 4.7 ± 2.1% of neurons in control cortices (n = 325 cells) (Fig. 4g), consistent with binucleation and with the known function of Cep55 in abscission.

In cultured cells, Cep55 knockdown results not only in binucleation but also in cells connected by intercellular bridges (ICBs) for a prolonged period of time[27,29]. We therefore stained brain sections from E13.5 samples with antibodies for Aurora B, an important constituent of the ICB, which has been previously used for identifying ICBs in the brain[39], and MKLP1 (also known

as KIF23) to visualize the MB. Interestingly, we observed that ICBs were present in similar numbers in control and *Cep55*-knockout cortical sections (Fig. 4h, i and Supplementary Fig. 4b). FLASH staining to visualize ICBs in intact cortices at E18.5 also showed no difference between control and *Cep55*-knockout mice (Supplementary Fig. 4c). Thus, *Cep55* knockout in the brain cortex does not result in a prolonged abscission stage in vivo that translates into a measurable accumulation of MB-bearing cells, but rather in some cells becoming binucleated.

**Cep55 is dispensable for cell division in most tissues.** Given Cep55's essential function in cell division in cultured cells, our finding that *Cep55*-knockout mice are born without severe gross morphological defects, except for the brain, is surprising. The lungs, liver, heart, and kidneys had similar weights in control and mutant newborns (Fig. 2b). However, we observed a slight but significant reduction in the body weight of mutant E18.5 embryos excluding the brain, compared with controls (Fig. 2b). FLASH analysis of whole E13.5 *Cep55*$^{-/-}$ embryos revealed C3A staining in the spinal cord region, particularly in the sensory ganglia, and in the cortex (Supplementary Fig. 3c). Conventional immunohistochemistry techniques confirmed that both central and peripheral nervous systems were apoptotic in mutant mice (Supplementary Fig. 5). *Cep55*$^{-/-}$ kidneys also showed small numbers of caspase 3-positive cells at E13.5 (Supplementary Fig. 5). However, most organs from mutant mice appeared normal, with proliferation and apoptosis similar to controls (Supplementary Figs. 5 and 6). The number of cells at the abscission stage was also comparable between control and *Cep55* knockout in several tissues at E13.5 and E18.5 (Supplementary Figs. 7 and 8). Thus, cell death by apoptosis in the developing nervous system is likely to be the main reason for the reduced tissue mass in embryonic and newborn *Cep55* mutant mice.

To determine whether Cep55 is required for cell division in the adult, we crossed *Cep55*$^{F/F}$ mice with a Villin-Cre line, to induce *Cep55* deletion in the intestinal epithelium (Supplementary Fig. 9a–c). These animals survived in good health through adulthood (Supplementary Fig. 9d). Conditional deletion of *Cep55* in the intestine, a highly proliferating tissue, did not affect intestinal integrity (Supplementary Fig. 9e). By using β-catenin immunostaining to visualize the membrane of intestinal cells[40], we found rare binucleated cells, at similar frequencies, in both the control and Cep55-depleted small intestine (Supplementary Fig. 9f–i). In addition, cell survival, assessed by C3A staining, and proliferation, assessed by phospho-H3 (PH3) staining, were not affected by *Cep55* deletion (Supplementary Fig. 9j–o). Thus, Cep55 is largely dispensable for completing cell division in the adult intestine and during embryogenesis.

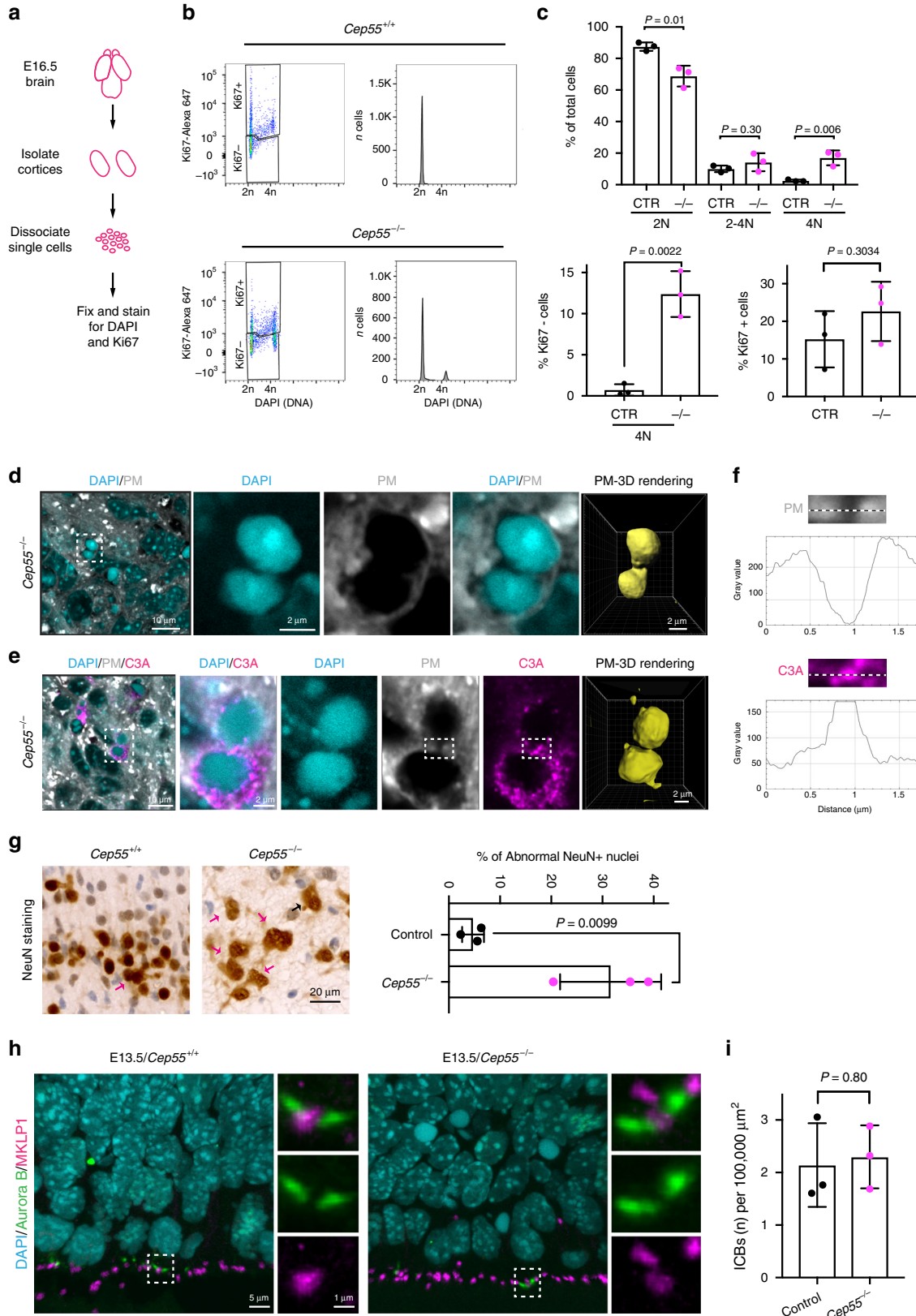

**Cep55 mediates abscission in NPCs but not in fibroblasts.** To examine the requirement for Cep55 in brain and body cells more closely, we established cultures of adherent embryonic neural progenitor cells (NPCs) and primary TTFs. As it was difficult to establish viable NPCs from *Cep55*−/− cortices, we turned to the conditional *Cep55^F* allele (Fig. 1a). Addition of 4-hydroxytamoxifen (4OHT) to the culture medium activates the Cre-ER(T2) present in these cells and resulted in efficient depletion of Cep55 in NPCs (Supplementary Fig. 10a–c and Methods). Immunofluorescence microscopy showed a significant

**Fig. 4 Cep55 deletion results in binucleated cells in the cortex. a** Scheme showing preparation of cells from E16.5 cortices for flow cytometry analysis. **b** Dot plots showing Ki67 staining and DNA content in $Cep55^{+/+}$ and $Cep55^{-/-}$ cortical cells. Histograms show DNA content of the Ki67-negative (Ki67−) cell populations in the lower boxes. **c** Quantifications from **b**. $n = 3$ and 6 mice per genotype, respectively; $n$ cells > 3000. CTR, control; $^{-/-}$, $Cep55^{-/-}$. P-values calculated using Student's two-tailed unpaired t-test. **d**, **e** Confocal optical section of E16.5 cortex from $Cep55^{-/-}$ brain stained with DAPI (DNA), plasma membrane (PM) stain, and active caspase 3 (C3A). Left boxed areas in **d** and **e** are magnified on the right. PM-3D rendering highlights connection of the two cells by a channel. **f** Top, enlargements of boxed areas from PM and C3A panels in **e**. Below, corresponding plots showing the fluorescence intensity along the dotted lines, in arbitrary units. Note that the PM appears interrupted. **g** Left: Immunohistochemistry of P0 $Cep55^{+/+}$ and $Cep55^{-/-}$ cortices for the neuronal marker NeuN. Magenta arrows, NeuN-positive doublets; black arrow, abnormally large nucleus. Right: Quantification of abnormal NeuN-positive (NeuN+) nuclei shown in **g**. Control includes $Cep55^{+/+}$ and $Cep55^{+/-}$ mice. $n = 3$ mice per genotype; $n = 325$ and 284 cells quantified, respectively. P-values calculated using Student's two-tailed unpaired t-test. **h** Confocal 3D images of E13.5 cortex from brains of the indicated genotypes stained with DAPI (DNA), Aurora B (green), and MKLP1 (Magenta). Boxed areas are magnified at right. **i** Quantification of intercellular bridges (ICBs) as shown in boxed areas in **h**. $n = 3$ mice per genotype; $n = 95$ and 59 ICBs, respectively. P-values calculated using Student's two-tailed unpaired t-test. All bar charts show mean ± SD. Source data for **c**, **g**, and **i** are provided as a Source Data file.

increase in binucleated and pyknotic cells, often present as doublets, in the resulting Cep55-deleted NPC cultures (+4OHT) compared with controls (Fig. 5a–c), consistent with our in vivo results.

In contrast to NPCs, $Cep55^{-/-}$ TTFs, which grew logarithmically in vitro, did not show any significant difference in the number of binucleated cells by immunofluorescence (Fig. 5d–f and Supplementary Fig. 11a). The proportion of TTFs at the abscission stages was also comparable between the two genotypes (Supplementary Fig. 11b). However, flow cytometry revealed a slight increase in cells with 4N DNA content, from 17.3 ± 1.6% (mean ± SD) in control cultures to 22.8 ± 1.5% in $Cep55^{-/-}$ cultures (Supplementary Fig. 11c, d). In addition, cells with >4N DNA content increased from 1.31 ± 0.06% (mean ± SD) in control to 2.02 ± 0.18% in Cep55-knockout cultures (Supplementary Fig. 11d), suggesting that some rare cells might have failed cell division. To test this hypothesis, we performed acute deletion of the $Cep55^F$ allele in MEFs (Supplementary Fig. 11e–j). We found a slight increase in binucleated cells from 9.8 ± 2.6% (mean ± SD) in control cultures (+EtOH) to 14.7 ± 2.1% in Cep55-depleted (+4OHT) cultures (Supplementary Fig. 11j). Thus, Cep55 depletion has a minor effect on cell division in primary fibroblasts in vitro, in contrast to the major effect observed in NPCs.

Next, we used live-cell imaging to directly monitor the completion of cell division in NPCs and TTFs. Inducible Cep55 deletion resulted in abscission failure in 69.33 ± 10.5% of NPCs ($n = 32$) compared with 15.67 ± 4.8% of controls ($n = 56$) (Fig. 5g, h and Supplementary Moviess 2 and 3). Long-term imaging of NPCs showed that some daughter cells that failed abscission stayed connected for several hours, up to 53 h in some cases, before detaching and dying (Supplementary Fig. 10d, e). Interestingly, Cep55 deletion also impaired the survival of non-dividing NPCs compared with control cultures (Supplementary Fig. 10d, e). In contrast, $Cep55^{-/-}$ TTFs formed an intercellular bridge and 78.9 ± 4.9% of cells ($n = 171$) successfully completed abscission compared with 73 ± 1.5% of control fibroblasts ($n = 153$; Fig. 5i, j and Supplementary Movies 4 and 5). A similar proportion of cells in control (19 ± 2.4%) and $Cep55^{-/-}$ TTF cultures (15.4 ± 3.8%) showed regression of the cleavage furrow before formation of the ICB and became binucleated (Binucleated/Furrow regression category in Fig.5i, j). Abscission failure (regression of the cleavage furrow after formation of the ICB) resulting in binucleated cells was observed at similar low frequencies in control (5.4 ± 1.3%) and mutant (1.7 ± 0.9%) TTFs (Fig. 5i, j). Importantly, mutant TTFs progressed with normal timing through abscission (Supplementary Fig. 11k).

As attachment to different substrates influences the ability of cells to complete abscission[41–43], we tested whether Cep55-knockout TTFs can complete abscission when cultured on dishes

coated with poly-L-lysine (PLL) and custom-made soft (0.5 kPa) and stiff (64 kPa) fibronectin matrices (Methods). $Cep55^{-/-}$ TTFs efficiently completed cell division under all these conditions (Supplementary Fig. 12a–d and Supplementary Movie 6). Although fibroblasts grew very poorly when plated on uncoated custom-made soft (0.2 kPa) plates, the majority of dividing cells completed cell division (Supplementary Fig. 12e). Together, these results indicate that Cep55 is essential for abscission in neural progenitors but dispensable in primary fibroblasts under different substrate conditions.

**Cep55-null fibroblasts divide in the absence of ESCRTs at the MB.** As Cep55 recruits the ESCRT complex at the MB to complete cell division[6,7], we investigated the localization of Cep55 and the ESCRT machinery by immunofluorescence, using α-tubulin to identify the ICB and MKLP1 for the MB. Cep55 was localized as expected in 93.5% of MBs from the mouse immortalized epithelial cell line EpH4 ($n = 93$) and in 65.6 ± 3.5% of MBs from wild-type NPCs ($n = 193$) (Fig. 6a, b). However, only 30.3 ± 2.9% of MBs from control TTFs ($n = 237$) showed some Cep55 signal, whereas the protein was absent at 97.9 ± 2.1% of MBs from $Cep55^{-/-}$ TTFs ($n = 214$) (Fig. 6a–e). Immunofluorescence analysis of the ESCRT-III core subunit Chmp2B showed the expected localization[6–9] at 88% of late stage MBs from EpH4 cells ($n = 73$), at 65.8 ± 11.3% of MBs from neural progenitors ($n = 90$), and only at 31.9 ± 6.1% of MBs from $Cep55^{+/+}$ fibroblasts ($n = 151$) (Fig. 6f–i). We could not detect any Chmp2B signal at 93.1 ± 0.2% of MBs from $Cep55^{-/-}$ cells ($n = 133$) (Fig. 6f–i). Chmp2B immunofluorescence analysis gave similar results when control and Cep55-knockout TTFs were grown on PLL-coated coverslips (Fig. 6f, j–l).

In immortalized fibroblasts stably expressing another ESCRT-III subunit, Chmp4B, tagged with enhanced green fluorescent protein (EGFP) (mChmp4B-EGFP) (Supplementary Fig. 13a–c), mChmp4B-EGFP localized at only 34 ± 8.2% of MBs from $Cep55^{+/+}$ cells ($n = 156$) (Supplementary Fig. 13d–f), and only if endogenous Cep55 was detectable (Supplementary Fig. 13g, h). Consistent with the Chmp2B staining, we also found that mChmp4B-EGFP was absent from 91.4 ± 8.8% of MBs of $Cep55^{-/-}$ immortalized fibroblasts ($n = 88$) (Supplementary Fig. 13d–f). Neither $Cep55^{+/+}$ nor $Cep55^{-/-}$ immortalized fibroblasts accumulated detectable EGFP-mALIX at the MB, in agreement with ALIX being dispensable for cell division in the mouse[22]. We detected EGFP-mTsg101 at the MBs of 21.5 ± 9.6% $Cep55^{+/+}$ fibroblasts ($n = 101$) but not in $Cep55^{-/-}$ immortalized fibroblasts (Supplementary Fig. 13i, j).

As ESCRT-III components are transiently recruited at the abscission site, it is possible that imaging of fixed cells may have missed the presence of ESCRT-III. To test this possibility, we

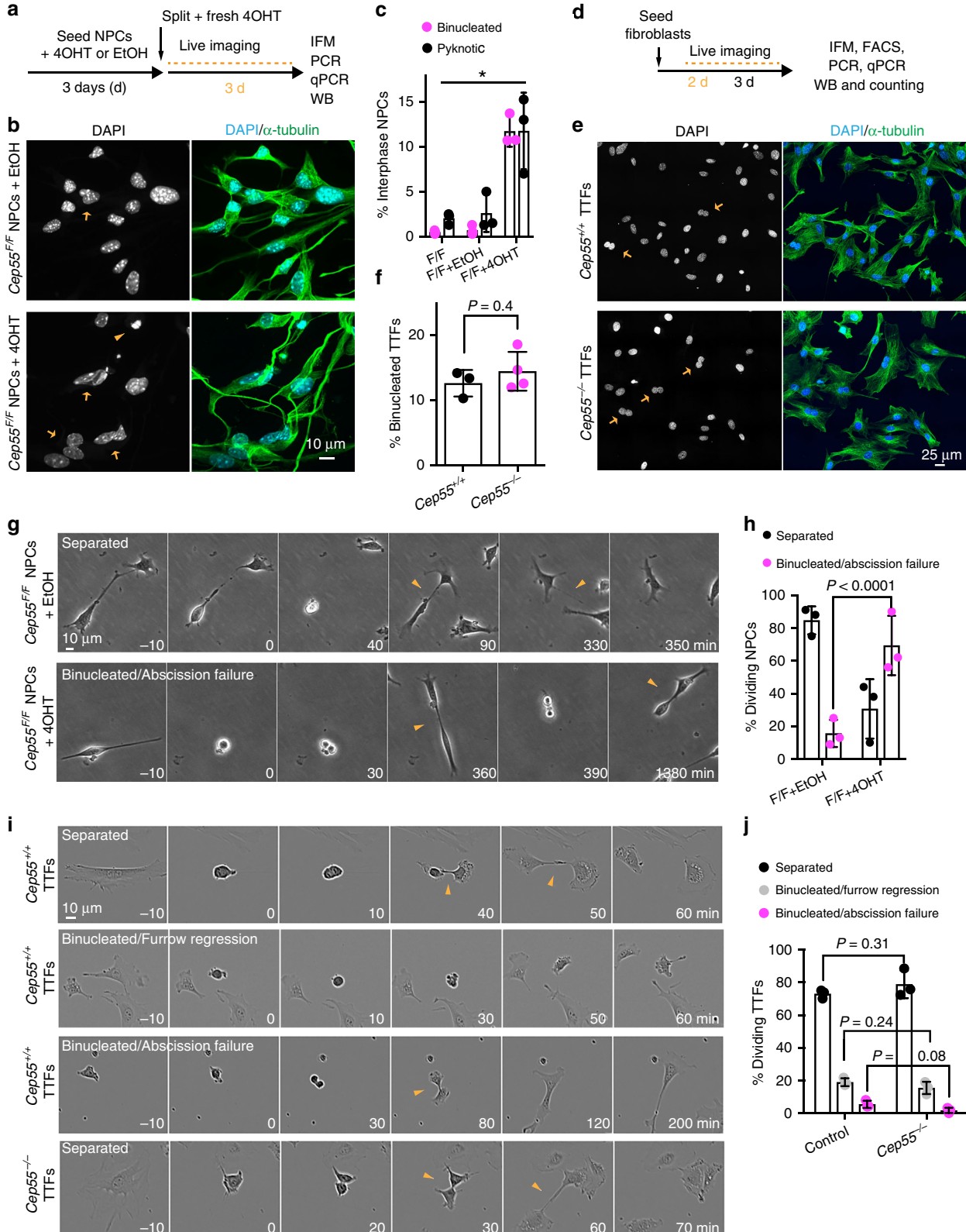

performed live-cell imaging of fibroblasts expressing mChmp4B-EGFP. Consistent with the immunofluorescence results, only 36% (n = 9/25) of $Cep55^{+/+}$ fibroblasts recruited mChmp4B-EGFP at the ICB before abscission (Fig. 6m, n). Thirty-two percent (n = 8/25) of $Cep55^{+/+}$ fibroblasts recruited mChmp4B-EGFP at the ICB only after abscission was completed and before the ICB

was resealed[44] (Fig. 6m, n). Thirty-two percent (n = 8/25) of $Cep55^{+/+}$ fibroblasts and 100% (n = 23/23) of $Cep55^{-/-}$ fibroblasts completed abscission without recruiting mChmp4B-EGFP either before or after abscission was completed (Fig. 6m, n). Similar results were obtained when TTFs were imaged on PLL-coated glass-bottom dishes (Fig. 6m, n and Supplementary

**Fig. 5 *Cep55* is required for abscission in neural progenitors but not in primary fibroblasts. a** Timeline for *Cep55* deletion in *Cep55$^{F/F}$; R26-Cre$^{ERT2/+}$* neural progenitor cells (NPCs), using 4-hydroxytamoxifen (4OHT) to induce recombination or ethanol (EtOH) as vehicle control. IFM, immunofluorescence microscopy; PCR, polymerase chain reaction; qPCR, quantitative PCR; WB, western blotting. **b** NPCs treated as shown in **a** were stained for DNA (DAPI) and α-tubulin. Arrows and arrowheads indicate binucleated and pyknotic cells, respectively, quantified in **c**. $n = 3$ mice per condition; 1000 cells quantified per condition. *P-values were calculated by one-way ANOVA followed by Tukey's multiple comparisons test as follows: binucleated cells: $P = 0.9703$ for FF vs. FF + EtOH, $P < 0.0001$ for FF vs. FF + 4OHT, $P < 0.0001$ for FF + EtOH vs. FF + 4OHT; pyknotic cells: $P = 0.9673$ for FF vs. FF + EtOH, $P = 0.0126$ for FF vs. FF + 4OHT, $P = 0.0165$ for FF + EtOH vs. FF + 4OHT. **d** Timeline for culture and analysis of mouse primary tail tip fibroblasts (TTFs). FACS, fluorescence-activated cell sorting. **e** TTFs of the indicated genotypes collected as in **d** were stained as in **b**. Arrows indicate binucleated cells, quantified in **f**. $n = 3$ and 4 mice, respectively; 1300 *Cep55$^{+/+}$* cells and 1800 *Cep55$^{-/-}$* cells quantified. **g** Representative time-lapse images of NPCs treated as in **a**. Arrowhead in the upper panel indicates the intercellular bridge; arrowhead in the lower panel indicates the attempt to divide possibly by cytofission[41]. **h** Quantification of dividing NPCs as defined in **g**. $n = 3$ mice per condition; 56 control cells (*Cep55$^{F/F}$* + EtOH) and 32 recombined cells (*Cep55$^{F/F}$* + 4OHT) were quantified. **i** Representative time-lapse images of TTFs cultured as in **d**. Arrowheads indicate the intercellular bridge. **j** Quantification of dividing *Cep55$^{+/+}$* and *Cep55$^{-/-}$* TTFs as defined in **i**. $n = 3$ mice per genotype; 153 and 171 cells quantified, respectively. All bar charts show mean ± SD. P-values calculated using Student's two-tailed unpaired t-test in **f**, **h**, **j**. *Cep55$^{F/F}$* indicates *Cep55$^{F/F}$; R26-Cre$^{ERT2/+}$* allele in **b**, **c**, **g**, **h**, *Cep55$^{-/-}$* indicates *Cep55$^{tm1a/tm1a}$* allele in **e** and **f**, and *Cep55$^{tm1d/tm1d}$* allele in **i** and **j**. Source data for **c**, **f**, **h**, **j** are provided as a Source Data file.

Movies 7–10). Altogether, these data suggest that, in the absence of Cep55, fibroblasts complete cell division without recruiting ALIX, Tsg101, or the ESCRT-III components Chmp2B and Chmp4B at the MB, and that the majority of wild-type fibroblasts divide without Cep55.

**Chmp4B mediates abscission in NPCs but not in fibroblasts.** To directly test whether cells can complete cell division in an ESCRT-independent way, we depleted several ESCRT-III subunits by RNA interference (RNAi) in neural progenitors and primary fibroblasts. In live-cell imaging experiments, knockdown of the essential cytokinesis protein MKLP1[45] in both *Cep55$^{+/+}$* and *Cep55$^{-/-}$* fibroblasts efficiently resulted in binucleated cells (Supplementary Fig. 14a–c). However, knockdown of the ESCRT-III subunits *Chmp2A*, *2B*, and *4B* did not affect the completion or timing of abscission in either control or *Cep55$^{-/-}$* primary fibroblasts (Supplementary Fig. 14b–d), although many cells died as previously reported[6,46]. As some ESCRT components might work redundantly, we depleted Chmp4B in *Chmp4C$^{-/-}$* primary fibroblasts (mice have no *Chmp4A* gene). Although we could not detect Chmp4C protein in *Chmp4C$^{+/+}$* TTFs by using a specific antibody, *Chmp4C* mRNA, detected with primers spanning exons that translate critical protein domains, was efficiently depleted (Supplementary Fig. 15). Abscission failure was observed at similar low frequencies in wild type ($5.9 \pm 0.6\%$, $n = 117$), *Chmp4C$^{-/-}$* ($3.1 \pm 2\%$, $n = 127$) and Chmp4B and C co-depleted fibroblasts ($1.7 \pm 1.7\%$, $n = 128$) (Fig. 7a–d and Supplementary Movies 11–13), and abscission timing was not affected in these cells (Supplementary Fig. 16). Chmp4B and C co-depleted fibroblasts could complete cell division in a similar manner when cultured on PLL covered dishes (Fig. 7a–d, Supplementary Movie 14, and Supplementary Fig. 16). In striking contrast, *Chmp4B* knockdown in NPCs resulted in failure of abscission in $65.2 \pm 10.1\%$ of cells ($n = 62$) compared with $6.3 \pm 2.1\%$ in controls ($n = 45$) (Supplementary Fig. 14a, Fig. 7e, f, and Supplementary Movies 15 and 16). Together, these data support the hypothesis that the Cep55-ESCRT pathway is dispensable for completing cell division in primary fibroblasts but essential in neural progenitors.

## Discussion

Successful cell division requires the orderly recruitment of the abscission machinery at the MB, to allow the severing of the intercellular bridge that connects daughter cells. In human cancer cell lines, the adaptor protein Cep55 is essential for the MB localization of ALIX and Tsg101, which in turn promote the assembly of the ESCRT-III abscission complex[6,7]. However, the role(s) of Cep55 in a mammalian organism remained unknown.

Here we show that Cep55 is important for cell division and survival of neural progenitors during the development of the nervous system, but dispensable for most cell division in the mouse (Fig. 7g). Remarkably, the majority of embryogenesis occurred normally in mice lacking *Cep55* and the rapid proliferation of the adult intestine was unaffected in *Cep55$^{F/F}$* conditional null mice. Our data also show normal intercellular bridges in intact *Cep55*-null organs and tissues as visualized by three-dimensional FLASH imaging. These findings suggest Cep55 as a regulator of abscission in specific contexts, similar to other MB proteins[47–53], rather than essential for abscission. It will be interesting to examine the requirement for Cep55 in other specialized cell divisions in the mouse, particularly in those, such as germ cells, where ESCRT-III is known to be crucial in other organisms[17,18].

The *Cep55$^{-/-}$* mouse phenotype of severe microcephaly, with diminished cellularity of the cerebral cortex and frequent binucleated neurons, is highly reminiscent of that observed in infants affected by MARCH syndrome[31], making it likely that the *Cep55* truncation mutations found in affected infants retain little if any Cep55 function. Our work provides insight into the etiology of this syndrome by showing that embryonic neural progenitors require Cep55 for survival and for completing abscission during cell division. It is notable that these two functions are not necessarily directly linked: in the *Cep55$^{-/-}$* mouse brain, most neural progenitors die as mononucleated cells, likely without undergoing cell division. Live-cell imaging of in vitro cultured *Cep55$^{F/F}$* neural progenitors confirms that more cells die while not dividing than during cell division (Supplementary Fig. 10d, e). Similarly, disruption of the MB protein citron kinase has independent effects on cytokinesis and apoptosis, specifically in the brain[54]. Cep55 has a pro-survival role during brain development in zebrafish, possibly through its ability to bind and stimulate the PI3K activity[55,56]. Because of the important role of PI3K in cytokinesis[57], it will be important to elucidate whether the pro-survival role of Cep55 is coupled with its role in cell division. Of note, apoptotic cells in *Cep55* mutant zebrafish are mononucleated, suggesting that Cep55's function in cell division might have evolved later, with brain expansion in mammals. Supporting this hypothesis, we found that Cep55 is most abundant at E13.5 in the nervous system and its depletion strongly reduces Tbr2-positive basal progenitors at this stage, when neurogenic divisions that expand the cortex are at their peak[38].

As well as brain abnormalities, diverse kidney defects have been reported for MARCH and Meckel-like syndromes[31–33], including true renal dysplasia, but also a range of other, less well-characterized changes. The rare *Cep55*-knockout mice that

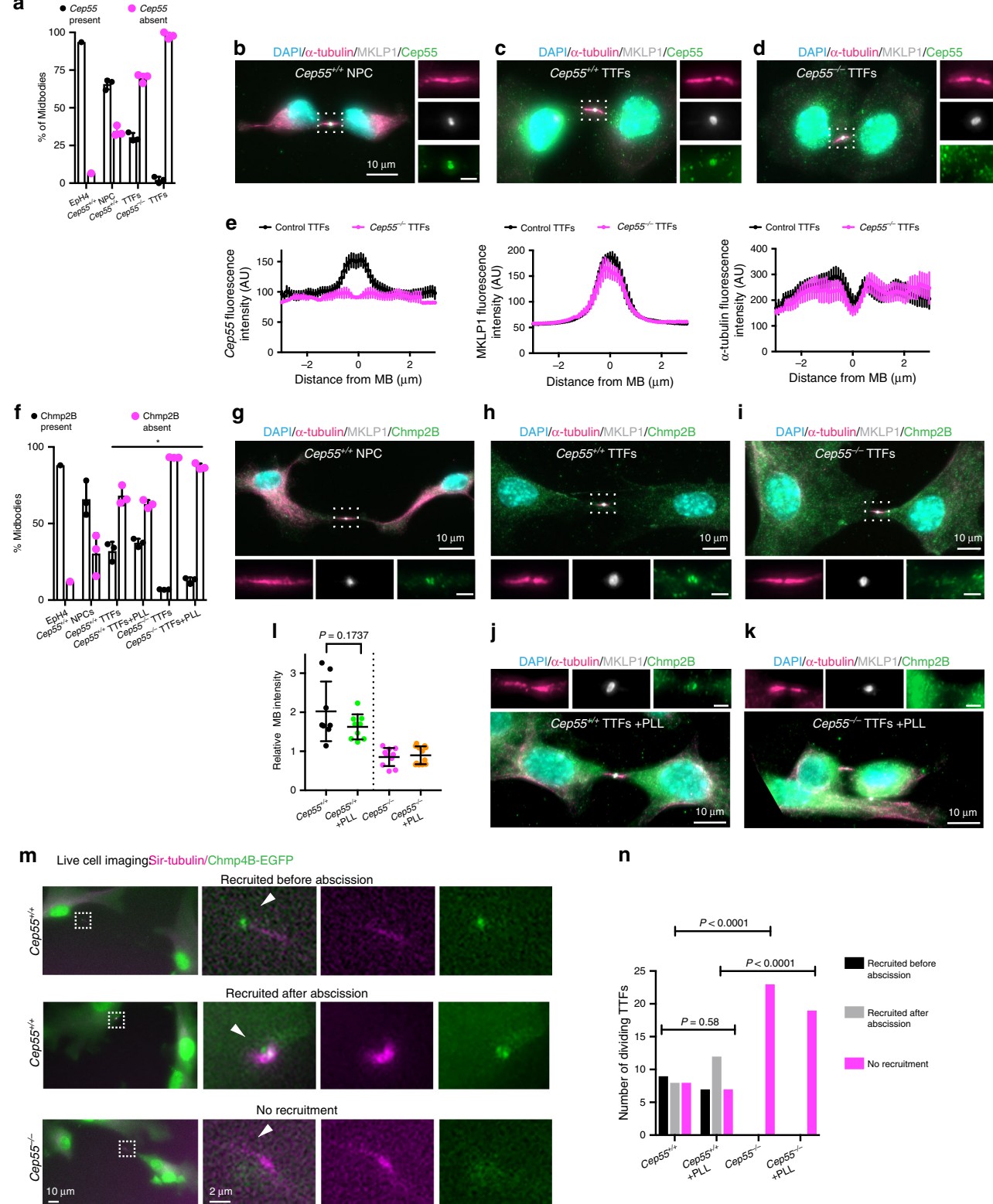

survived at 2 weeks of age demonstrated kidney lesions typical of dysmaturity and hypoplasia, rather than renal dysplasia. Although we found a few, possibly binucleated, apoptotic cells in the kidneys of E13.5 mutant mice, we could not detect major differences in size, proliferation, or cells at the abscission stage in the kidneys of E13.5, E18.5, and newborn mice. In contrast, *Cep55* mutant zebrafish show renal atrophy during development[31]. Similarly, disruption of the MB protein Kif14 is associated with microcephaly and kidney defects in human and zebrafish, but only with microcephaly in the mouse[48,52,53]. It is therefore possible that the changes observed in the kidney of 2-week-old mutant mice are mainly due to impaired postnatal, rather than embryonic, development. The extensive apoptosis observed in the nervous system is likely to account for the rapid

**Fig. 6 ESCRTs are recruited at the MB of neural progenitors but are absent in *Cep55*-knockout fibroblasts. a** Quantification (mean ± SD) of MBs with Cep55 present or absent in the indicated cell types. $n = 3$ mice per cell type; 193, 237, and 214 cells quantified, respectively. **b–d** Immunofluorescence images of intercellular bridges in the indicated cell types stained for DNA (DAPI), α-tubulin, MKLP1, and Cep55. Boxed areas are magnified on the right. Scale bar in magnified regions, 2 μm. Punctate cytoplasmic signal is nonspecific staining. **e** Plots showing the fluorescence intensity along the intercellular bridge, in arbitrary units (AU), from TTFs as in **c** and **d**. Ten cells per genotype quantified. SEM are shown. **f** Quantification (mean ± SD) of MBs with Chmp2B present or absent in the indicated cell types. $n = 3$ mice per cell type; 90, 151, 133, 86, and 67 cells quantified. *P*-values for TTFs were calculated using two-way ANOVA, each mean compared with control *Cep55*$^{+/+}$ (plastic), followed by Dunnett's multiple comparisons test. For "Chmp2B present" category, *P*-values are: 0.2057, 0.0001, and 0.0004, respectively. **g–k** Immunofluorescence images of the indicated cell types undergoing abscission stained for DNA (DAPI), α-tubulin, MKLP1, and Chmp2B. Boxed areas are magnified below (**g–i**) or above (**j, k**). Scale bar, 2 μm. **l** Quantification of Chmp2B signal (mean ± SD) in TTFs grown on glass or poly-L-lysine (PLL)-coated glass coverslips as in **h–k**. For *Cep55*$^{+/+}$ TTFs, only Chmp2B-positive cells were analyzed. $n = 8, 9, 10$, and 12 cells analyzed. *P*-value calculated using Student's two-tailed unpaired *t*-test. **m** Time-lapse images of control and *Cep55*$^{-/-}$ TTFs expressing mChmp4B-EGFP and stained with Sir-tubulin to visualize the microtubules of the ICB. The white arrowheads indicate the site of abscission on the intercellular bridge. **n** Quantification of mChmp4B-EGFP at the intercellular bridge as in **m**. Individual pairwise comparisons were performed using $\chi^2$-test (one-sided). The omnibus $\chi^2$-test *P*-value is < 0.0001. $n = 25, 26, 23$, and 19 cells were imaged. Source data for **a**, **e**, **f**, **l**, and **n** are provided as a Source Data file.

postnatal death of *Cep55* mutant mice, although kidney dysfunction might contribute.

When cultured in vitro, neuronal progenitors required not only Cep55 but also ESCRT-III to complete abscission, in agreement with the known function of Cep55 in recruiting ESCRT-III. In contrast, *Cep55*-knockout primary fibroblasts could not recruit ALIX, Tsg101, or the ESCRT-III components Chmp2B and 4B at the MB. Nevertheless, they could undergo normal cytokinetic abscission even after depletion of key ESCRT-III components. Importantly, most wild-type fibroblasts divided in the absence of detectable Cep55 at the MB. It is possible that the level of ESCRT activity required for division is simply much lower in fibroblasts, below the level of residual protein remaining after knockdown and below the threshold of detection by immunofluorescence. However, it is not clear how such a mechanism would work in the absence of Cep55. In *Drosophila*, which lack Cep55, MKLP1 (known as Pavarotti) can recruit ALIX or Tsg101 that in turn recruit Chmp4B (known as Shrub)[17,58,59]. As none of these proteins, except MKLP1, are recruited in our mouse TTFs in the absence of Cep55, any alternative ESCRT recruitment mechanism must be ALIX and Tsg101 independent.

One factor potentially affecting the different abscission requirements in fibroblasts and neural progenitors is the relative stiffness of the originating tissues, which would enable greater exertion of traction forces in fibroblasts compared with NPCs in vivo. Traction forces at the cell-substrate interface have been proposed to promote abscission in fibroblasts[41–43,60], whereas neural progenitors have been reported to proliferate maximally on low-stiffness substrates[61]. MBs from neural epithelial cells and HeLa cells have a very similar lipid composition[62,63], and altered plasma membrane lipid composition during abscission in HeLa has been suggested to increase mechanical resistance to rupture[62]. It is possible that in such cell types, severing of the intracellular bridge requires the assistance of Cep55 and ESCRT, whereas these proteins may not be necessary in cell types in which traction forces are sufficient to break the plasma membrane. However, we found that *Cep55*$^{-/-}$ mouse primary fibroblasts formed a normal ICB and were able to divide successfully even when cultured on low-stiffness substrates (Supplementary Fig. 12e), indicating that mechanical cues are not essential even in the absence of Cep55. It remains to be explored how such cells divide in a more physiological environment.

In conclusion, our study highlights that the Cep55-ESCRT pathway is essential for faithful division of NPCs, similar to some cancer cell lines in vitro, whereas abscission in most other tissues does not require Cep55. The full requirement for ESCRT-III in vivo remains to be explored, and will require extensive experimentation using a range of approaches to fully dissect the

pleotropic and possibly compensatory roles of the different complex components. If ESCRT-III is dispensable in vivo, it would mean that a Cep55-ESCRT-independent mechanism of abscission likely drives cell division in most of the cells of a mammalian body. *Cep55*$^{-/-}$ cells may aid in identifying this mechanism, which would address a major unresolved question in our understanding of cell division.

## Methods

**Mouse lines.** All experiments were approved by the Animal Welfare and Ethical Review Body of the Francis Crick Institute, and conformed to UK Home Office regulations under the Animals (Scientific Procedures) Act 1986 including Amendment Regulations 2012.

For the generation of *Cep55* mice, ES cell clone HEPD0726_6_A04 carrying the knockout first allele *tm1a* was obtained from the European Conditional Mouse Mutagenesis (EUCOMM) Program[34]. The targeted ES cell clone (agouti C57BL/6 parental cell line JM8A3.N1) was injected into blastocysts of C57BL/6 background to generate chimeric mice, which were then crossed with wild-type C57BL/6 mice for germline transmission. The conditional *tm1c* and knockout *tm1d Cep55* mice were generated by crossing with *Flpo* and *Pgk1*-expressing mice, respectively (Fig. 1a). The *Flpo* and *Pgk1* alleles were bred out and mice were maintained on a C57BL/6 background. Sperm from *Chmp4C-tm1a* mice was obtained from EUCOMM (strain name EPD0113_1_B10) and used for in vitro fertilization of oocytes from wild-type C57BL/6 females to obtain *Chmp4c-tm1a* heterozygous mice. Heterozygous mice were intercrossed and generated viable *Chmp4c*$^{-/-}$ offspring, as previously reported[64]. *Rosa26CreER(T2)* mice[65] and *Villin-Cre* transgenic mice[66] have been described.

**Genotyping.** Genotyping was performed by Transnetyx (https://www.transnetyx.com/home) using real-time PCR. DNA was extracted with Direct PCR reagent (Viagen 102-T) and used with Taq DNA polymerase (Qiagen 201205) according to the manufacturer's instructions. Genotype primer sequences are provided in Supplementary Fig. 17.

**Cell culture and treatments.** Primary mouse tail fibroblasts (TTFs) were generated from E18.5 and newborn mouse tail tips by finely mincing 1–2 mm tissue and placing it in growth medium. MEFs were derived from E13.5 embryos, from which the head and internal organs had been dissected out, by mincing and placing them in growth medium. Primary and immortalized TTFs, MEFs and EpH4 cells were cultured in DMEM medium (GIBCO 41966-029) supplemented with 10% fetal bovine serum (GIBCO 10500064) and 100 U/ml penicillin/streptomycin (GIBCO 15140-122). In all experiments, except where indicated, TTFs were grown on Corning® Costar® TC-Treated Multiple Well Plates (3516 and 3526) and Corning® tissue-culture-treated culture dishes (430167 and 430599). Corning® Costar® TC-Treated Multiple Well Plates and CytoSoft® 6-Well Plates (0.2–64 kPa) (Advanced BioMatrix 5190-7EA), were treated overnight with PLL (SIGMA P4832) at 4 °C and for 2 h with 40 μg/ml fibronectin solution (SIGMA F1141) at room temperature (RT), respectively. SiR-tubulin (Cytoskeleton CY-SC002) was used at 50 nM in imaging medium (ThermoFisher, A1896701).

NPCs were isolated from E13.5 cortices and cultured as neurospheres in RHB-A medium (Takara Y40001) complemented with human epidermal growth factor (20 ng/ml; PeproTech) and fibroblast growth factor (FGF-basic; 20 ng/ml; PeproTech). Neurospheres were dissociated with Accumax (Sigma A7089). Adherent NPC cultures, which were used throughout this study, were obtained by treating the growth surface of the culture vessels for 2 h with a 100 μg/ml Poly-D-lysine solution (Sigma P7280) at RT followed by an overnight treatment with a 15 μg/ml laminin

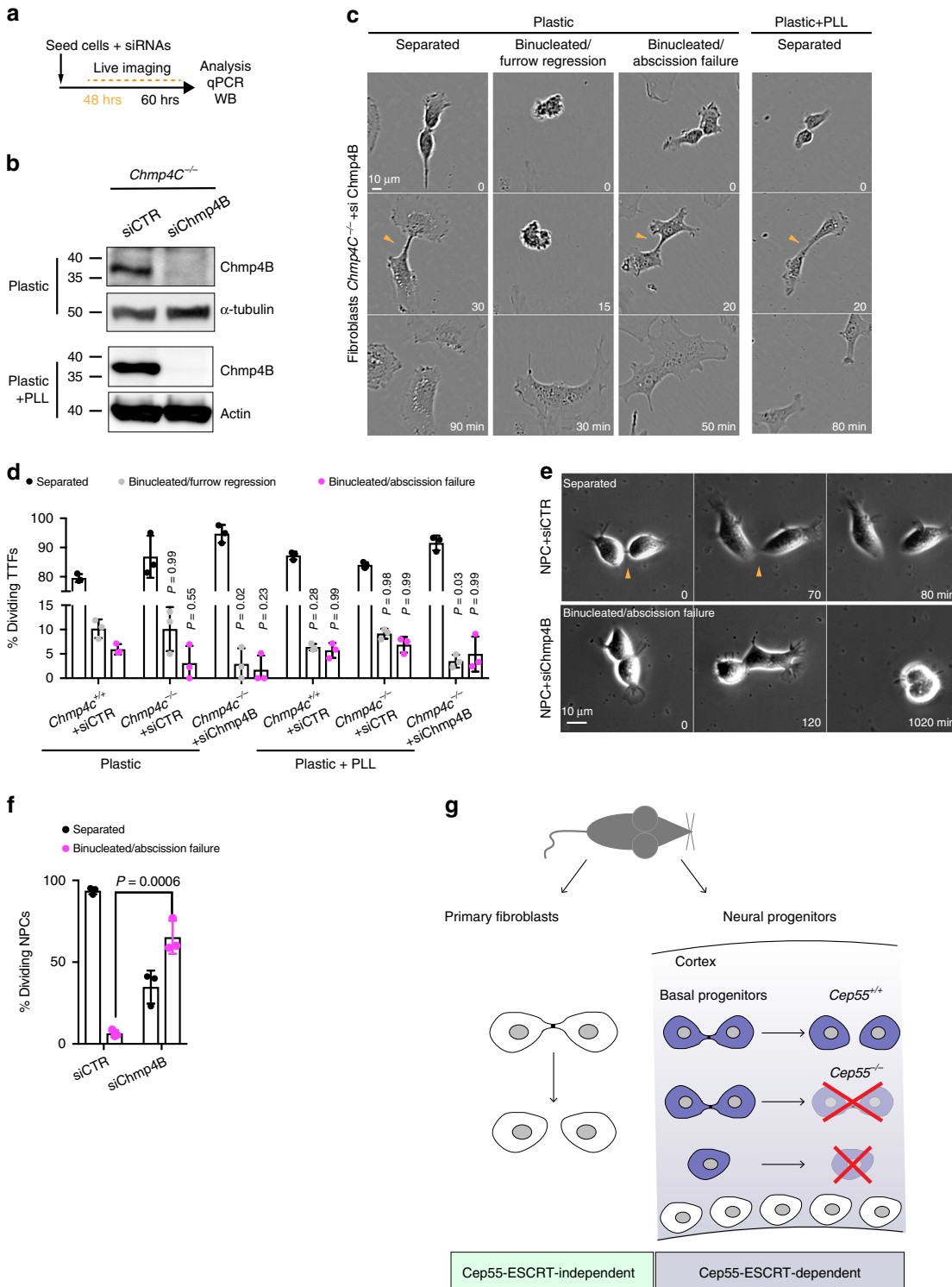

**Fig. 7 ESCRTs are required for abscission in neural progenitors but not in primary fibroblasts. a** Scheme of Chmp4B knockdown in *Chmp4C*$^{-/-}$ fibroblasts and wild-type NPCs. siRNA, small interfering RNA. **b** *Chmp4C*$^{-/-}$ fibroblasts were transfected with the indicated siRNAs for 60 h as in **a** and extracts analyzed for Chmp4B and α-tubulin or actin. CTR, control; si, small interfering RNA; $n = 1$ and 2 independent experiments, respectively. **c** Time-lapse images of *Chmp4C*$^{-/-}$ fibroblasts depleted of Chmp4B as shown in **a**. Arrowheads indicate the intercellular bridge. **d** Quantification of abscission success or failure in dividing fibroblasts, as defined in **c**. $n = 3$ mice per condition; 117, 127, 128, 125, 163, and 115 cells quantified, respectively. *P*-values were calculated using two-way ANOVA, each mean compared with *Chmp4C*$^{+/+}$ on plastic, followed by Dunnett's multiple comparisons test. **e** Time-lapse images of wild-type NPCs depleted of Chmp4B as shown in **a**. **f** Quantification of abscission success or failure in dividing NPCs, as defined in **e**. $n = 3$ mice per condition; 62 and 45 cells quantified respectively. *P*-values calculated using Student's two-tailed unpaired *t*-test. **g** Model of the different abscission requirements in vivo. Red crosses indicate dying cells. All bar charts show mean ± SD. Source data for **b**, **d**, and **f** are provided as a Source Data file.

solution (Sigma L2020) at 4 °C. Adherent NPCs for live-cell imaging were grown on glass-bottom dishes (MatTek 10564).

Conditional deletion of the $Cep55^F$ allele in MEFs (Supplementary Fig. 11e–j) and NPCs (Fig. 5a–c, g, h and Supplementary Fig. 10) was achieved by adding 0.5 and 0.25 µM 4OHT (Sigma), respectively. All experiments with primary cells were performed between passages two and five. All cell types were grown at 37 °C, 5% $CO_2$ and 5% $O_2$.

**Stable cell lines.** cDNAs for mouse (m) Chmp4B (NM_029362.3), Tsg101 (NM_021884.4), and ALIX (NM_001164677.1) were amplified from mouse normal tissue RNA (Qiagen) using Takara's PrimeScript High Fidelity RT-PCR Kit (R022A), the required restriction sites at the 5′- and 3′-ends added, and the inserts sub-cloned into the appropriate pBabe (puromycin) EGFP 13-glycine linker or pBabe C-13 glycine linker EGFP receptor plasmid to generate mChmp4B-EGFP, EGFP-mTsg101, and EGFP-mAlix. DNA primer sequences are provided in Supplementary Fig. 17. Tags were added to N- or C-termini of the mouse genes as in the published human versions[6,8] and molecular weights matched those observed for the respective human equivalents. All plasmids were made by MRCPPU Reagents and Services (http://mrcppureagents.dundee.ac.uk) at the University of Dundee. Retroviral supernatants were collected 48 hours after transfection of the above constructs in Phoenix-Eco cells with Polyethylenimine (Polysciences 23966-2) according to the manufacturer's instructions. $Cep55^{+/+}$ and $Cep55^{-/-}$ tail fibroblasts were immortalized with a short hairpin RNA against p16 (blasticidin selection, Sigma, 5 µg/ml) and the resulting cells were transduced with the above supernatants and selected with puromycin (Gibco, 2 µg/ml).

**Histopathology, immunohistochemistry, and immunocytochemistry.** Histo-pathological phenotypic analysis was performed by A. Suárez-Bonnet and S. L. Priestnall, two board-certified veterinary pathologists from the Royal Veterinary College (UK) providing support at the Francis Crick Institute. Organs (brain, skin, intestine, spleen, kidney, pancreas, liver, heart, lung, and thymus) were isolated from two controls and three Cep55-knockout mice, all of which were 2 weeks old (example kidneys shown in Supplementary Fig. 2) and stained for HE. All examined tissues, with the exception of the brain and kidneys, although subjectively smaller in size/volume, were microscopically unremarkable with no observable difference from control mice.

Mouse brains from E13.5, E16.5, and P0 mice, and gut rolls, were dissected in cold phosphate-buffered saline (PBS) 1× (Gibco 14190-094). All specimens were fixed in 10% (vol/vol) neutral buffered formalin for 24 hours, transferred to 70% (vol/vol) ethanol, processed and embedded into paraffin. Sections were cut (sagittally for brain sections) at 2 and 4 µm for HE staining, 3,3′-diaminobenzidine staining, and fluorescence immunohistochemistry.

For cell counting, all cells in the CP, IZ, SVZ, and VZ of the brain cortex were counted in an area of the same width. Pax6 staining was used to determine the extent of the VZ (e.g., Fig. 3i). C3A (Supplementary Fig. 5) and Ki67 (Supplementary Fig. 6) positive cells were automatically counted with QuPath software[67].

Fibroblasts and adherent neural progenitors were grown on 13 mm glass coverslips (Menzel), fixed with 4% paraformaldehyde for 10 min or, when staining for MB proteins, with 100% Methanol for 16 h and stained according to standard immunofluorescence methods. For immunohistochemistry and immunofluorescence, antibodies to C3A (R&D AF835, 1:900), Ki67 (Abcam 16667, 1:350), NeuN (Chemicon MAB377, 1:600), Tbr2 (Abcam 23345, 1:1500), Pax6 (DSHB, 1:250), Cep55 (Santa Cruz 377044, 1:10), MKLP1 (Santa Cruz 136473, 1:200 or Abcam 170344, 1:1451), α-tubulin (Sigma B512, 1:6000 or Abcam 6161, 1:500), Chmp2B (Bethyl 304501, 1:50), GFP (Abcam 6673, 1:100), Aurora B (BD 611082) CellMask™ Orange membrane stain (ThermoFisher, 1:1000), and Alexa fluorophore (488, 568, or 633)-conjugated antibodies (Molecular Probes, 1:1000) were used. DNA was counterstained with DAPI (Roche Diagnostics) at 2 µg/ml or DRAQ5 (INVITROGEN 65-0880-96) at 10 µM for FLASH. Slides were mounted using Vectashield Mounting medium (Vector Laboratories).

**FLASH.** Whole E13.5 embryos were processed for FLASH[36] by replacing Borate-SDS with a solution of 3-[dimethyl(tetradecyl)azaniumyl]propane-1-sulfonate detergent (80 g/L), Boric acid (200 mM) and Urea (250 g/L). Brain, heart, lungs, liver, intestine, and kidneys were micro-dissected from E18.5 embryos and pieces of tissue from comparable regions between control and Cep55 knockout were processed for FLASH as above.

**Microscopy and image analysis.** Images in Fig. 2d, g, 3a, b, d-f, i, k, 4g and Supplementary Figs. 2, 3d, 5, 6 and 9 were acquired on Zeiss Axio Scan Z1 Slide Scanner with a ×20 objective and Supplementary Fig. 7 with a ×40 objective controlled by Zen 2.3 software. Images for Fig. 4d–f were acquired on a Zeiss Upright 710 laser confocal scanning microscope using a Plan Apochromat x63/1.4 NA oil objective lens (Zeiss) controlled by Zen 2010 software and shown as single confocal planes. The three-dimensional (3D) rendering images were obtained with Imaris 8.3.1 software. Images in Fig. 4h and Supplementary Figs. 4 and 8 were acquired with a Zeiss 780 inverted confocal microscope using a Plan Apochromat ×40/1.3 NA oil objective lens. Images in Fig. 5b, e were acquired on a spinning disk

confocal microscope CV1000 Yokogawa, fitted with a Photometrics Evolve 512 × 512 electron modifying charge-coupled device camera. Supplementary Fig. 3c shows 3D rendering images acquired in light-sheet LaVision UltraMicroscope II.

Images for Fig. 6b–d, g–i, k, l and Supplementary Fig. 13d, e, g, i were acquired on a Zeiss Axio Imager M1 microscope using a Plan Apochromat ×63/1.4 NA oil objective lens (Zeiss) equipped with an ORCA-Spark CMOS camera (Hamamatsu) and controlled by Micro-Manager 2.0 software[68]. Images were processed with Fiji/ImageJ 1.4 and 1.5[69], and displayed as maximum-intensity projections of Z planes that were acquired in 200 nm sections. To quantify the level of recruitment of ESCRT proteins to the MB, maximum projections of z-stacks of MBs were analyzed. The central MB area was identified using MKLP1 localization and the intercellular bridge (ICB) was identified by combining the MKLP1 channel with microtubule staining; the MB and ICB were found using auto-thresholding methods with objects selected by size and circularity, ICB regions flanking the central MB were identified by subtracting the central MB area from the ICB area. Chmp2B intensity was then measured in both the central MB area and outer areas, ratio of mean intensity at MB/outer areas indicating recruitment (conditions with no recruitment, having a ratio ~1).

Long-term imaging in Fig. 5g, i, Fig. 7c, and Supplementary Figs. 10, 12 and 14 was performed using an Incucyte FLR device equipped with a 20x objective (Essen BioScience). Live-cell imaging in Fig. 7e was performed on a Nikon Eclipse Ti2 equipped with a chamber at 37 °C and 5% $CO_2$ using a Plan Apochromat ×20/075NA Ph2 objective.

Live-cell imaging in Fig. 6m was performed using a Nikon Ti2 inverted microscope with Perfect Focus System and ASI motorized XY stage with Piezo Z, using a Plan Apochromat 40×/0.95 Ph2 objective, Okolab environmental chamber at 37 °C and 5% $CO_2$ and a Prime BSI scientific CMOS camera (Photometrics). The microscope was controlled with Micro-Manager v2.0 software (Open-Imaging). Fluorescence excitation was performed using a SpectraX LED light engine (Lumencor) fitted with standard excitation filters: 470/24 nm for Chmp4B-EGFP and 640/30 nm for SiR-Tubulin-647. Emission filters were ET - EGFP single-band bandpass filter ET525/50 nm (Chroma) and 680/42 nm BrightLine® single-band bandpass filter (Semrock), with a 409/493/573/652 nm BrightLine® quad-edge dichroic beamsplitter. To image cytokinesis, Z-stacks (9 × 1 µm step) were acquired every 3 min.

**Abscission stages and timing.** As ESCRTs are recruited at the MB in mid-late abscission stages[8,9], we excluded early abscission stages, identified by the presence of thick microtubule bundles in the intercellular bridge, from quantifications in Fig. 6f–l and Supplementary Fig. 13.

Intercellular bridges were identified on two-dimensional sections using Aurora B and MKLP1 localization (Fig. 4h and Supplementary Figs. 4a and 7) and in 3D tissue samples using Aurora B and α-tubulin (Supplementary Figs. 4b and 8).

Abscission duration (Supplementary Figs. 11k, 12b, d, 14d, and 16b) was taken as the time period between the formation of two distinct daughter cells and the resolution of the intercellular bridge (see e.g. Fig. 7c, e).

**Western blotting.** Cells were lysed on ice with Laemmli sample buffer supplemented with protease inhibitor cocktail (SIGMA), PMSF (SIGMA), and β-mercaptoethanol (SIGMA). Lysates were sonicated, heated to 95 °C for 5 min, separated by SDS-polyacrylamide gel electrophoresis (precast gels, Biorad), and transferred with a trans-Blot Turbo transfer system (Biorad). Antibodies against α-tubulin (Sigma B512, 1:10,000), Cep55 (Cell Signaling 81693S, 1:1000), Chmp4B (Abcam 105767, 1:250), Chmp4C (Abcam 155668, 1:1000), GAPDH (Abcam 9485, 1:1000), GFP (Abcam 6673, 1:200), Actin-HRP (Abcam, ab4900, 1:5000), and anti-HRP (Jackson, 1:5000) were used according to standard western blotting methods. Chemiluminescence was performed with normal ECL (GE Healthcare) and blots were imaged with Amersham imager 600.

**RNA interference.** The following small interfering RNAs were used at a final concentration of 25 nM: SMARTpool Dharmacon: Chmp2a (M-045840-01), Chmp2b (M-044691-01), Chmp4b (M-041531-01), MKLP1 (M-05710-00), and Non-targeting Pool (D-001206-14). Transfection was performed with Lipofectamine RNAi MAX (Invitrogen) according to the manufacturer's instructions. As indicated in Fig. 7a, transfection reactions were added into the growth medium immediately after seeding the cells.

**Flow cytometry.** Cells from E16.5 cortices were dissociated using the papain Dissociation Kit (Worthington Biochemical Corporation) according to the manufacturer's instructions. Single cell suspensions were fixed with 70% ethanol for 1 h on ice. A minimum of $2 \times 10^5$ cells per condition were stained with DAPI and Alexa Fluor 647 anti-mouse Ki-67 Antibody (BioLegend 652407) according to the manufacturer's instructions. For propidium iodide staining, logarithmically growing tail fibroblasts and MEFs from a 10 cm dish were suspended in 800 µl of PBS and added to 2.2 ml of ice-cold methanol for 1 h on ice. Fixed cells were resuspended and incubated in 1 ml of 50 µg/ml propidium iodide, 10 mM Tris pH 7.5, 5 mM $MgCl_2$, and 200 µg /ml RNase A for at least 30 min at 37 °C in the dark. Fluorescence was measured with a BD LSRII (BD Biosciences, San Jose) flow cytometer and data were analyzed with FlowJo (9 or above) software.

**Uncropped blots and flow cytometric analysis gating**. All uncropped and unprocessed blot scans are provided in the Source Data file. Flow cytometric analysis gating information is shown in Supplementary Figs. 4a and 11i.

**Quantitative real-time PCR analysis**. RNA was extracted with RNeasy Mini Kit (Qiagen, 74104) according to the manufacturer's instructions. qRT-PCR analysis was performed with iSCRIPT cDNA Kit (Biorad, 1708891) reagents and Power up SYBR Green master mix (Applied Biosystems, 100029284) according to the manufacturer's instructions. qRT-PCR primer sequences are provided in Supplementary Fig. 17.

**Statistics and reproducibility**. Statistical analyses were carried out using the GraphPad Prism software package (https://www.graphpad.com/). $P$-values were calculated using unpaired two-tailed Student's $t$-test (for single comparisons of two samples), one- or two-way analysis of variance followed by Dunnett's or Tukey's multiple comparisons test (for more than one comparison), or by Fisher's exact test or $\chi^2$-test (for categorical variables), as noted in the figure legends. The exact sample sizes ($n$) used to calculate statistics are provided in the figure legends. $P$ values are provided in the figure panels or, if not possible, in the legends. All experiments were reproduced with similar results a minimum of two times, and the exact number of repetitions is provided in the figure legends.

**Reporting summary**. Further information on research design is available in the Nature Research Reporting Summary linked to this article.

## Data availability

All relevant data are included with the manuscript or available from the authors upon request. The source data underlying Fig. 1e, d, 2b, c, e, f, h, 3c, g, h, j, l, 4c, g, i, 5c, f, h, j, 6a, e, f, l, n and 7b, d, f and Supplementary Figs 1d, c, 2b, e, 3b, c, e, 4c, 5c, 6d, 7e, 8f, 9b, c, i, l, o, 10b, c, e, 11a, b, d, g-k, 12a-d, 13f, h, j, 14a, b, d, 15d, e, and 16a, b are provided as a Source Data file.

## Code availability

The code to analyze the data in Fig. 6j is freely available from https://github.com/FrancisCrickInstitute/CALM

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

## Acknowledgements

This work was supported by the Francis Crick Institute, which receives its core funding from Cancer Research UK (FC001039), the UK Medical Research Council (FC001039), and the Wellcome Trust (FC001039). We are grateful to the Biological Research, Experimental Histopathology, Flow Cytometry, Genomics Equipment Park, and Light Microscopy facilities at the Francis Crick Institute. We thank F. Uhlmann, J. Carlton, T. Takaki, J. Nelson, G. Koifman, and C. Cremona for comments on the manuscript, G. Kelly for help with the statistical analysis, C. Gribben for help with stainings, L. Chmelova for technical assistance, and G. Stamp, A. Suárez-Bonnet, and S. L. Priestnall for histopathology analysis.

## Author contributions

Conceptualization: A.T., M.P., A.B. Methodology, formal analysis, investigation, writing (original draft): A.T. Methodology and investigation (FLASH): J.A., and H.A.M. Investigation and analysis (fluorescence live-cell imaging and code): M.A.R. Writing (review and editing): A.T., M.P., A.B. Funding acquisition: M.P. and A.B.

## Competing interests

H.A.M. and A.B. are inventors on a UK patent application (1818567.8) relating to a solution for the preparation of samples for 3D imaging. The remaining authors declare no competing interests.
