## [Peer Review File · Nature Communications]

Reviewers' comments:

Reviewer #1 (Remarks to the Author):

In the present manuscript, Tedeschi et al. analyze the consequences of knocking out in mice Cep55, a protein known to play a key role for recruiting the ESCRT machinery at the midbody and thus for cytokinetic abscission in cultured cells. The authors first showed that Cep55 KO lead to mice with impaired motor coordination and microcephaly. Cytokinetic defects (binucleated cells) and cell death were found specifically in the brain, affecting neural progenitors in vivo. In vitro, Cep55^{-/-} neural progenitors also displayed delayed abscission and a proportion of binucleated cells. In contrast, Cep55^{-/-} fibroblasts divided normally. Consistently, the ESCRT machinery was recruited in a Cep55-dependent manner at the midbody in neural progenitors, whereas none of the tested ESCRT components were routinely found in wild-type fibroblast cultured in vitro. In addition, depletion of ESCRT components in fibroblasts had no effect on cell separation in vitro. The authors propose that both Cep55 and the ESCRT machinery are dispensable for most cell division in mice, except for neural progenitors.

Such conclusions are both provocative and exciting. As such, this manuscript should receive a strong interest from the cell division community. However, a number of conclusions are based on assumptions that are questionable: 1) In vivo, Cep55 KO equals absence of ESCRTs during cytokinesis and 2) normal abscission in vitro (e.g. upon Cep55 or ESCRT depletion) equals normal abscission in vivo. Thus, in my opinion, several important issues should be addressed before this manuscript can be published in Nature Communications.

Major points

1. The manuscript would greatly benefit from a more substantial introduction and discussion, in order to describe the state of the art regarding the role of Cep55 and ESCRT-III in cytokinesis.

For instance, there is no Cep55 in *Drosophila* whereas ESCRT-III is essential for normal stem cell cytokinesis (PMID 25635693 and 25647097). Actually, the fact that Cep55 is dispensable for most cell division in mammals is not entirely new and was suggested by human genetics (ref. 25-26-27), as stated in the discussion. It is also known that ESCRT-III is not essential for somatic cytokinesis in *Drosophila* (PMID 25635693 and 25647097) and in *C. elegans* (PMID 28325808). Finally, abscission eventually occurs, although it is strongly delayed, in human cells in culture and sometimes leads to binucleation (PMID 26929449).

2. The claim that the ESCRT machinery is dispensable for cytokinesis has not been demonstrated in vivo. Unless there is evidence, the title, the abstract and the text should be revised accordingly. This would need to KO ESCRT-III components in vivo, presumably in a conditional manner and to analyze the effects on abscission timing or binucleation. Indeed, Cep55 KO does not guarantee that ESCRT-III is not recruited in vivo. In *Drosophila* cells, Shrub is localized at the midbody and is required for abscission, yet there is no Cep55 (PMID 24217622). Thus alternative mechanisms for ESCRT machinery recruitment exist, at least in other organisms.

3. The fact that binucleated cells are not observed in most tissues upon Cep55 KO does not necessarily mean that Cep55 is dispensable for cytokinesis in vivo. One can imagine that binucleated cells arise in Cep55^{-/-} cells but are efficiently eliminated in most tissues in a p53-dependent manner (PMID 16222300). In this respect, Cep55 KO in a p53^{-/-} background would be very informative. Otherwise, the authors should revise their conclusions. Alternatively, Cep55 deletion might only delay abscission without formation of binucleated cells. In this case, the authors should at least analyze in tissue sections or explants whether Cep55 KO leads to an increase of cells connected by tubulin/MKLP1-positive bridges in vivo. Abscission tests using diffusion of photo-activated GFP would be ideal but difficult to set up in vivo.

4. It is hazardous to infer that the ESCRT machinery is dispensable in vivo in most tissues based on in vitro studies. It is already known that abscission can occur in vitro in the absence of detectable CEP55 or ESCRT-III by cytofission, when the cells can migrate on artificial environments like glass (see PMID 29507669). This paper should be acknowledged and discussed in the light of the present results. This raises important questions. Do Cep55^{-/-} or ESCRT-depleted fibroblasts fail cytokinesis when cultured on poly-L-lysine or soft environments (see PMID 29507669)? What is the ESCRT localization in these conditions? In addition, the authors should analyze the ESCRT localization in cells in their natural environment, in wt tissues sections or explants if they want to conclude that ESCRTs are not present in cytokinetic bridges in vivo.

Minor points

1. In which cells is Cep55 expressed in vivo?

2. The abstract and title should be revised to avoid overstatements (see points above). For instance ESCRT-III is essential for the normal timing of abscission in cultured cells, but is not essential for cell division as stated.

3. Supplementary Fig. 1: what is the expected size of the Cep55 protein in Cep55^{-/-}?

In panel 1d, I cannot see the difference in Cep55 +/+ vs. Cep55-/-.

4. Are there binucleated cells in highly proliferative tissues such as the intestine? If the binucleated cells were eliminated, are there more mitotic cells to compensate?

5. When abscission timing is measured, the cumulative plots should be provided in supplementary data (not only means and SDs) for proper comparisons. Duration of abscission in Fig. 3j and 5d would be interesting.

6. In video 3, the presented cell does not fail to complete abscission, as stated. Instead, the cell quickly becomes binucleated and later tries to separate the two nuclei by cytofission. This has been described in PMID 23878225. There is thus no cytokinetic bridge.

7. Videos of GFP-CHMP4b in Eph4 cells and in wt fibroblasts should be provided. GFP-CHMP4b might be present at the midbody in all cells, but only transiently and before abscission. Thus, results on fixed cells might give the false impression that CHMP4B is usually not recruited in wt fibroblasts. This could also depend on culture conditions (see above see PMID 29507669).

8. Discussion: KIF14 mutations also lead to microcephaly and could be discussed as well

(e.g. PMID 28892560 and 30388224). There are also frequent kidney defects and ciliogenesis defects in KIF14 or Citron Kinase mutants. Is it the case in Cep55 mice? If not analyzed, the authors should be cautious not to conclude that there is no defect in kidneys.

9. Typo p.8 line 3: Fig. 5e,f

Reviewer #2 (Remarks to the Author):

The authors report the surprising result that proliferation and development of most murine tissues occurs reasonably normally in the absence of the ESCRT adaptor, Cep55. Neuronal progenitor cells are the one clear exception, which provides a nice demonstration that the ESCRT pathway is important for abscission in some contexts. These observations are extended to primary cells in culture, where primary fibroblast abscission is convincingly shown to occur in the absence of Cep55, and upon efficient depletion of several key ESCRT-III factors, whereas neuronal progenitors (and murine cell lines) are shown to require Cep55 and the ESCRT pathway for efficient abscission.

Evaluation

This is a high-quality study that reports a surprising result and adds significantly to the growing body of evidence that ESCRT-mediated abscission is less important *in vivo* than in transformed cell lines in culture. This basic result is important, convincing and should be of general interest to the cell biology community. The experiments also appear to be well performed, well controlled and quite thorough. I therefore do not have suggestions for additional required experiments. My only comment of substance is that I think the authors should be a bit more cautious in entirely dismissing abscission roles for the ESCRT pathway in different contexts. The reported experiments makes it clear that cells in most living tissues can divide without ESCRT recruitment, which is an important advance, but I think it would be surprising if Cep55/ESCRT do not make contributions in any cell types beyond neural progenitors. For example, *Tex14*, which competitively inhibits the ability of Cep55 to recruit TSG101 and ALIX, is required for stable intercellular bridge formation during germ cell formation (<https://www.ncbi.nlm.nih.gov/pubmed/19020301> and references therein). Moreover, it looks to me as though there may be room for small effects even in fibroblasts because the cultured primary Cep55^{+/+} fibroblasts appear to replicate slightly faster than their Cep55^{-/-} counterparts (Supplemental Figure 3c), although this effect may or may not be significant because the Cep55^{+/+} fibroblasts are also slower in this experiment. Similarly, 4OHT treatment appears to increase the number of binucleated Cep55 F/F MEFs slightly (Supplemental Figure 3j). None of this is meant to suggest that the results being reported are not significant or surprising, just to suggest that the Discussion (which is generally well written) might be modified slightly to accommodate these possibilities.

Reviewer #3 (Remarks to the Author):

The manuscript submitted by Dr. Tedeschi and co-Authors addresses the *in vivo* function of Cep55 in mouse development. In contrast with a supposedly generalized function of Cep55 in cytokinesis completion, they show that this protein is mostly required for brain development. Cep55 loss results in microcephaly and reduced weight at birth, but embryo morphogenesis occurs quite normally. Neural progenitors show abnormal cytokinesis and elevated levels of apoptosis. In contrast, embryo fibroblasts divide normally. Authors also show that, besides Cep55, even other ESCRTIII components are not required for cytokinesis of primary fibroblasts. The results are consistent with the phenotype produced in humans by CEP55 loss of function mutations.

Overall, this is a very interesting paper, supporting the view that cytokinesis of neural progenitors is regulated differently from other cell types, as well as that many proteins thought to be generally important for cytokinesis are, in fact, dispensable in most cases.

I have no major issues to raise but, in my opinion, addressing the following points would make the paper stronger:

1. The histological panels of Fig. 1 (e and h) could be significantly improved. The ko section appears significantly more lateral than control, giving the impression of a telencephalic reduction much higher than quantified. For this reason, even the higher power fields are not really comparable.

2. Since a significant number of mice seems to survive to P21, it would be nice to see para-sagittal brain sections at P14. This would allow a better description of which brain regions are most affected by Cep55 loss.

3. End of page 4, beginning of 5: I think the authors mean cortical plate, not preplate. It would be nice showing at least an example of doublets within a high magnification field of the cortex.

4. Since mice are smaller at birth, some defects must be present also in other cell types. I would strongly suggest a more careful analysis of embryo sections stained for C3A since, to my eye, it seems that some apoptotic cells are present, in different districts. The organization of kidney does not look exactly identical to control. This is quite relevant, since kidney alterations are observed in patients. Flow cytometry analysis of other tissues could also be helpful, to reveal eventual cell cycle abnormalities. In any case, Authors should at least discuss the possible reasons that make ko animals smaller at birth.

5. It would be nice showing in vivo protein loss, in addition to decreased RNA in the KO. In theory, this could be easily done by WB of developing brain. Added to the supplementary Fig. 1d, it would also increase the confidence that Cep55 is really undetectable in fibroblasts.

6. Authors are very careful in not stating that apoptosis is a consequence of cytokinesis failure, although they show that it frequently occurs in doublets. I suspect cell death could be at least partially not dependent from cytokinesis failure. However, since they have time lapse recordings of knockout divisions, it would be very nice to assess whether cell death occurs in non dividing cells, before anaphase or only after cleavage furrow ingression. In any case, I think they should at least comment in the discussion about the causal relationships between cytokinesis failure and apoptosis.

7. Apoptosis is most abundant in the intermediate zone, raising the possibility that it may occur in neurons. Authors may want to consider this in their model.

8. The title of Fig. 4 is not fully justified. In wild type fibroblasts, ESCRT is not absent, but can be detected in approximately 25% of midbodies. The signal is specific, since there is a clear difference with the knockout. For this reason, the title should be smoothed.

Ferdinando Di Cunto

Reviewer #4 (Remarks to the Author):

Tedeschi et al show here that Cep55 knockout neonates present with microcephaly but without significant impairment of other tissues. Lack of Cep55 leads to apoptotic and binucleated cortical cells, due to a requirement for Cep55 in neural progenitor cell abscission. Additionally they show that Chmp2b and Chmp4b are recruited to the midbody of neural progenitors, and that knockdown of multiple ESCRT-III components leads to abscission failure in neural progenitors but not primary fibroblasts.

While the data are interesting, in showing a specific requirement for Cep55 in cortical development through abscission of neural progenitor cells, there are several major and minor issues with the manuscript as listed below that I think would need to be carefully addressed before the manuscript is suitable for publication. Additionally, the novelty of this paper is somewhat compromised by the 2017 publication from Laporte et al, showing that Alix is required during development for normal growth of the mouse brain.

Major concerns:

Specifically the lack of proof of knockdown at the protein level throughout the paper.

Throughout the manuscript all stats have been performed using unpaired t-tests. This is not appropriate when more than 2 sets of data are present on any graph, as occurs in numerous figures. Appropriate statistical tests (one or two way ANOVAs) need to be performed for the appropriate graphs.

The discussion needs substantial expansion, and reference therein should be made to the above mentioned Laporte paper.

Major Points:

1. The validation of Cep55 knockdown in the mouse model is not complete. Although knockdown is shown at the mRNA level in Supplementary Figure 1 b, Supplementary Figure 1 d does not prove absence of Cep55 protein in fibroblasts. This blot has a very high background making lanes 3-7 extremely unclear, but as far I can see no Cep55 is visible in +/+ or +/- lanes. Furthermore, knockdown should be shown at the protein level in multiple tissues, as several different tissues are examined during this study (Supplementary Figure 2). This is of course extremely pertinent for brain tissues.

Additionally, proof of knockdown at the protein level would be more appropriately shown in Figure 1, with the uncropped blots shown in full in the supplemental to prove no protein truncation is occurring.

2. "Together, these data indicate that Cep55 deletion promotes microcephaly as result of cell death by apoptosis." From the representative image (Fig 2d) it looks like the cleaved caspase 3 staining seems to be restricted largely to the upper SVZ and IZ. This should be noted in the text. Has this been quantified?

3. Patients with Cep55 mutations present with brain and kidney defects. Supplementary Figure 2 d claims that HE staining of kidney shows no defect, however the tissue distribution looks different between genotypes in this representative image. Please amend or comment on this phenomenon.

As per major point 1, Cep55 depletion in these investigated tissues has not been verified.

4. Supplementary Figure 3 a,b: again, protein blots to quantify knockdown are needed here, as this is not a full knockdown at the mRNA level (Supplementary figure 3b- appropriate stats need to be performed and included).

5. Figure 4 d shows bright cytosolic punctate Cep55 staining in Cep^{-/-} fibroblasts. What is this?

6. Fig 3 c does not show a significant increase in pyknotic cells in treated cells compared to vehicle control ($p = 0.298$), although the text states there is a significant increase. Is this a typo on the graph?

7. Chmp4C^{-/-} fibroblasts. Methods state these KO mice were generated for this study. However no validation of the model is presented here. This model needs to be validated and appropriate data showing Chmp4c specific knockdown needs to be presented.

8. Fig 5 d and f.

In Fig 5 d it looks like there may be a significant difference between Chmp4c^{+/+} + Ctr siRNA and Chmp4c^{-/-} + Chmp4b siRNA. Is there? All significant differences should be stated.

Additionally, I'm confused by the inclusion of 2 different criteria for abscission failure in Fig 5 d, and only 1 abscission failure for 5 f. It looks like if the binucleated data in 5 d were pooled similarly to 5 f they might be significant. Please comment.

Minor Points:

1. Pg 4: "Cep55 was highly expressed in the developing central nervous system (Supplementary Fig. 1e,f), consistent with an essential role in brain development." Only P0 is shown in this figure supporting Cep55 expression in neonates, so this sentence needs to be rephrased or normal levels of Cep55 expression investigated at different timepoints of CNS development.

2. Figure 2 a and d: Although it is apparent that the layers are thinner, it appears that the correct layering of the cortex is intact. This should be noted in the text.

3. “Numbers of Ki67-positive nuclei scored by immunohistochemistry were similar in control and Cep55^{-/-} cortices (control 17.5±2.1% versus Cep55^{-/-} 21±1.7% at E16.5; n = 3470 and 2237 cells respectively, from 3 mice per genotype), suggesting that differences in proliferation rates do not explain the loss of cells in Cep55^{-/-} mice.” This should say percentage instead of numbers; the Cep55^{-/-} brains already have fewer cells (as evidenced by the number of cells per genotype). This should be linked to Fig 2g, which is omitted in the text. Why are these graphs and the stats not shown? The y axes in Fig 2g are not visible, the resolution needs to be improved.

4. “In newborn Cep55^{-/-} mice, 32.2±8.9% of 5 neurons in the cortical preplate layer (mean ±sd; n = 284 cells from 3 mice) also appeared as doublets or were abnormally large compared with 4.7± 2.1% of neurons in control cortices (n = 325 cells), consistent with binucleation and with the known function of Cep55 in abscission.” Where is this data? Or data not shown?

5. Fig 2J Distance (inches) is not appropriate when the scale bar on the figure from which these inserts are lifted is 2 μm.

6. Pg 5. I’m confused by the change to a conditional knockout at this point. Presumably this is because established full knockout does not produce culturable neural progenitors. If so, please state. Additionally the changes between conditional knockout and Cep^{-/-} fibroblasts from the mouse are confusing here, and could do with clarifying in the text.

7. “Supplementary Figure 3: Cep55 is dispensable for proliferation and division of primary fibroblasts.” This title needs to be amended as it includes data from conditional knockout NPCs as well as primary fibroblasts. Please clarify.

8. Pg 7 “and that the majority of wild-type fibroblasts divide without Cep55 and without detectable ESCRT-III.” However as has been investigated in the next section, ESCRT-III redundancy has not been investigated here, and presence of Chmp2a or Chmp4c cannot be excluded.

9. Fig 5 g The representative images for Pax6 staining look quite different between Cep^{+/+} and Cep^{-/-}, although the quantification does not support this. Please choose different representative images.

10. Fig 5 i For the Tbr2 staining, where the number of stained cells are significantly different at E13.5, it looks like there are more positively stained cells in the uppermost layer. Is this real?

Point-by-point response to reviewer comments for NCOMMS-19-10086-T 260619 by Tedeschi et al.

We thank the reviewers for their thorough and constructive criticism and suggestions for our work. The manuscript has now expanded considerably, with the addition of 2 main figures and 11 supplementary figures of new data to address the points raised. The reviewer comments are in black below, and our point-by-point responses in blue text.

Reviewer #1 (Remarks to the Author):

In the present manuscript, Tedeschi et al. analyze the consequences of knocking out in mice Cep55, a protein known to play a key role for recruiting the ESCRT machinery at the midbody and thus for cytokinetic abscission in cultured cells. The authors first showed that Cep55 KO lead to mice with impaired motor coordination and microcephaly. Cytokinetic defects (binucleated cells) and cell death were found specifically in the brain, affecting neural progenitors in vivo. In vitro, Cep55^{-/-} neural progenitors also displayed delayed abscission and a proportion of binucleated cells. In contrast, Cep55^{-/-} fibroblasts divided normally. Consistently, the ESCRT machinery was recruited in a Cep55-dependent manner at the midbody in neural progenitors, whereas none of the tested ESCRT components were routinely found in wild-type fibroblast cultured in vitro. In addition, depletion of ESCRT components in fibroblasts had no effect on cell separation in vitro. The authors propose that both Cep55 and the ESCRT machinery are dispensable for most cell division in mice, except for neural progenitors.

Such conclusions are both provocative and exciting. As such, this manuscript should receive a strong interest from the cell division community. However, a number of conclusions are based on assumptions that are questionable: 1) In vivo, Cep55 KO equals absence of ESCRTs during cytokinesis and 2) normal abscission in vitro (e.g. upon Cep55 or ESCRT depletion) equals normal abscission in vivo. Thus, in my opinion, several important issues should be addressed before this manuscript can be published in Nature Communications.

We thank the reviewer for the positive evaluation.

Major points

1. The manuscript would greatly benefit from a more substantial introduction and discussion, in order to describe the state of the art regarding the role of Cep55 and ESCRT-III in cytokinesis.

For instance, there is no Cep55 in Drosophila whereas ESCRT-III is essential for normal stem cell cytokinesis (PMID 25635693 and 25647097). Actually, the fact that Cep55 is dispensable for most cell division in mammals is not entirely new and was suggested by human genetics (ref. 25-26-27), as stated in the discussion.

We agree with the reviewer and have expanded both introduction and discussion accordingly to include data from different models and species, including the

references cited above (now ref. 20 and 21). The human genetics papers (now ref. 31-32-33) described the phenotype resulting from Cep55 truncation; our study now adds the phenotype of a Cep55 null, allowing stronger conclusions to be made regarding the requirement for Cep55 in mammalian cell division. Our findings now clarify that the Cep55 truncation mutations described in humans likely result in a complete loss of function. Moreover, our study additionally explores whether the ESCRTs are recruited at the midbody of primary cells in the absence of Cep55.

It is also known that ESCRT-III is not essential for somatic cytokinesis in *Drosophila* (PMID 25635693 and 25647097) and in *C. elegans* (PMID 28325808).

The previous studies mentioned revealed the importance of ESCRT-III in specific contexts (now ref. 20, 21 and 22). PMID 25635693 showed that ALIX KO flies are infertile because ALIX is required to recruit the ESCRT-III protein Shrub during cytokinesis of germline stem cells in *Drosophila*. PMID 25647097 specifically knocked down Shrub in the fly germ line and PMID 28325808 addressed the role of ESCRT-III during the first embryonic abscission in *C. elegans*. Our study additionally underlines the importance of ESCRT-III in neurogenesis in the mouse. As discussed below (major points 2 and 4), the role for ESCRT-III in somatic cytokinesis *in vivo* remains to be rigorously tested.

Finally, abscission eventually occurs, although it is strongly delayed, in human cells in culture and sometimes leads to binucleation (PMID 26929449).

The results in PMID 26929449 (now ref. 18), which reports ~15% of binucleated cells after ALIX knockdown in human HeLa cells, contrast with those in Refs. 5,6,14,16, which report >60% of binucleated cells and major binucleation for the knockdown of other ESCRT components. Regarding the abscission delay they and others report in HeLa, we do not find such a delay in cultures of primary fibroblasts depleted for ESCRT components (see minor point 5 below), underlining the need to distinguish the requirements of different cell types.

2. The claim that the ESCRT machinery is dispensable for cytokinesis has not been demonstrated *in vivo*. Unless there is evidence, the title, the abstract and the text should be revised accordingly. This would need to KO ESCRT-III components *in vivo*, presumably in a conditional manner and to analyze the effects on abscission timing or binucleation. Indeed, Cep55 KO does not guarantee that ESCRT-III is not recruited *in vivo*. In *Drosophila* cells, Shrub is localized at the midbody and is required for abscission, yet there is no Cep55 (PMID 24217622). Thus alternative mechanisms for ESCRT machinery recruitment exist, at least in other organisms.

We agree with the reviewer on this point and have now revised the title and text to discuss the possibility that a Cep55-independent mechanism might exist to recruit ESCRT-III at the midbody, as in *Drosophila* (PMID 25635693, ref 20). It has recently been shown that in *Drosophila* MKLP1 directly recruits ALIX in order to recruit Shrub at the MB, but the MKLP-ALIX interaction does not occur in human HeLa cells (PMID 31607533, ref 59). If a Cep55-independent mechanism exists in the mouse, it might be different from the one in *Drosophila* as ALIX is not essential in the mouse (PMID 28322231, ref 25) and we show that neither ALIX or Tsg101 is recruited at the midbody in Cep55 knockout fibroblasts (Supplementary Fig. 13j). As discussed in the

text (p.3), knockout of ESCRT components in vivo, even conditionally, is unlikely to be informative due to the multiplicity of ESCRT functions and component proteins. We have therefore amended the article title to: "Cep55 promotes cytokinesis of neural progenitors but is dispensable for most mammalian cell divisions", and have revised the abstract accordingly.

3. The fact that binucleated cells are not observed in most tissues upon Cep55 KO does not necessary mean that Cep55 is dispensable for cytokinesis in vivo. One can imagine that binucleated cells arise in Cep55^{-/-} cells but are efficiently eliminated in most tissues in a p53-dependent manner (PMID 16222300). In this respect, Cep55 KO in a p53^{-/-} background would be very informative. Otherwise, the authors should revise their conclusions.

Our conclusion that Cep55 is dispensable for most cytokinesis in vivo is based on the development and birth of live young that lack Cep55 protein. In order to complete embryogenesis, we would argue that thousands of successful cell divisions must have occurred in the absence of Cep55. Knockout of other key cell division factors results in early embryonic lethality, for example Ect2 and mgcRacGAP knockout blastocysts show binucleation and do not develop beyond E3.5 (Van de Putte et al, 2001, PMID 11287179; Cook et al, 2011, PMID 22701760) and Plk1 knockouts die at the 8-cell stage (Lu et al, 2008, PMID 18794363). It is of course possible that occasional binucleated cells arise in all tissues but are then eliminated from all tissues except the brain, and the reviewer is correct that a p53^{-/-} Cep55 KO mouse would determine whether this occurs in a p53-dependent manner. In the case of knockout of the ESCRT-I protein Tsg101, concomitant deletion of p53 resulted in a slight shift in the embryonic lethality from E6.5 to E8.5 (Ref. 23). As an alternative, we have carefully examined the possibility of increased apoptosis coupled with compensatory proliferation (please see our response to Reviewer #1, minor point 4, below). Two veterinary pathologists have now examined caspase 3 stained sections from E13.5 embryos. In addition to apoptosis in the nervous system, they have noted more frequent apoptotic cells in the mouth and in the kidneys (Supplementary Note and Supplementary Fig. 5). To further examine the possibility of apoptotic elimination of cells in all tissues, we have performed 3D imaging of the entire embryo (Supplementary Fig. 3c). This analysis confirmed an increase in apoptosis in the nervous system. However, the majority of tissues show no differences compared with controls (Supplementary Figs. 5 and 6 and Supplementary Note), supporting the interpretation that Cep55 is dispensable for most cytokinesis in vivo.

Alternatively, Cep55 deletion might only delay abscission without formation of binucleated cells. In this case, the authors should at least analyze in tissue sections or explants whether Cep55 KO leads to an increase of cells connected by tubulin/MKLP1-positive bridges in vivo. Abscission tests using diffusion of photo-activated GFP would be ideal but difficult to set up in vivo.

We agree that this is a possible scenario and have tried to experimentally clarify this point by examining abscission bridges in vivo. We have used FLASH, a technique that has been recently developed in the lab to analyse the 3D architecture of mouse tissues (ref 36). The advantage of FLASH is that we can examine abscission bridges in 3D rather than in 2D sections. By using FLASH, the MKLP1 antibody does not

give a signal, while Aurora B, which has been used for visualizing and calculating the abscission index in mouse brain cortex (ref. 39), gives a clear signal in brain cortex, lung, heart, liver and kidney at E18.5 enabling visualization of intercellular bridges in intact tissue (Supplementary Fig. 4b, Supplementary Fig. 8). We have also analysed conventional tissue sections of brain cortex, lung, heart, liver and kidney stained for Aurora B/MKLP1 at E13.5 (Figure 4h, Supplementary Fig. 4a and Supplementary Fig 7). Both analyses showed that Cep55 KO does not result in more cells connected by intercellular bridges in vivo, suggesting that abscission is not delayed to a degree that would translate into detectable readout in this assay.

4. It is hazardous to infer that the ESCRT machinery is dispensable in vivo in most tissues based on in vitro studies. It is already known that abscission can occur in vitro in the absence of detectable CEP55 or ESCRT-III by cytofission, when the cells can migrate on artificial environments like glass (see PMID 29507669). This paper should be acknowledged and discussed in the light of the present results. This raises important questions. Do Cep55^{-/-} or ESCRT-depleted fibroblasts fail cytokinesis when cultured on poly-L-lysine or soft environments (see PMID 29507669)? What is the ESCRT localization in these conditions? In addition, the authors should analyze the ESCRT localization in cells in their natural environment, in wt tissues sections or explants if they want to conclude that ESCRTs are not present in cytokinetic bridges in vivo.

We agree with the reviewer and have revised the title and abstract to focus on Cep55, as discussed in our response to Major point 2 above. The mentioned paper (PMID 29507669) has been acknowledged and discussed at the appropriate points throughout the paper (ref. 43). When grown on poly-L-lysine or soft substrates, Cep55 KO fibroblasts and ESCRT-III depleted fibroblasts successfully divide (Fig. 7c, d, Supplementary Fig. 12 and 16) and ESCRT localization and recruitment is not affected by culturing on PLL (Fig. 6f-I). These findings are discussed on pages 11-12 and 14.

Regarding the ESCRT localization at abscission bridges in vivo, we cannot detect any localization of Chmp2A, B and Chmp4B at the midbody in the brain, kidney, lung, heart, or liver at E13.5 after testing several antibodies. Since we cannot exclude that either these antibodies are not suitable for IF on sections or that we missed the stages when ESCRT proteins are recruited at the midbody, we have now removed the conclusion that the ESCRT machinery is dispensable for abscission in vivo. This point is discussed further on page 17.

Minor points

1. In which cells is Cep55 expressed in vivo?

LacZ staining at E13.5 (Supplementary Fig. 3a, previously Supplementary Fig. 1e) gives a strong signal in the CNS, consistent with the brain phenotype we describe here. We have tried 5 different Cep55 antibodies for immunohistochemistry but none gave a specific signal. We have now performed Cep55 WB, with a newly identified antibody (see Methods), on micro-dissected cortices from different developmental stages (Fig. 2c and Supplementary Fig. 3b) as well as micro-dissected lungs, liver,

gut, heart and kidneys from E19.5 embryos (Supplementary Fig. 1f). We found that Cep55 is most abundant in E13.5 cortices, but is present in all tissues examined.

2. The abstract and title should be revised to avoid overstatements (see points above). For instance ESCRT-III is essential for the normal timing of abscission in cultured cells, but is not essential for cell division as stated.

We have now revised the abstract and title as suggested.

3. Supplementary Fig. 1: what is the expected size of the Cep55 protein in Cep55 $-/-$? In panel 1d, I cannot see the difference in Cep55 $+/+$ vs. Cep55 $-/-$.

The expected size of mouse Cep55 is predicted to be close to a relative molecular weight of 55 kDa. We have now repeated the blot in previous Supplementary Figure 1d (now Fig 1e) with a new antibody, and a clear band around 50 kDa is detectable in control fibroblasts but not in Cep55 knockout fibroblasts.

4. Are there binucleated cells in highly proliferative tissues such as the intestine? If the binucleated cells were eliminated, are there more mitotic cells to compensate?

We thank the reviewer for this pertinent question. We have analysed sections from control and Cep55 depleted intestine after immunostaining for beta-catenin, to visualize the plasma membrane and detect binucleated cells, active caspase 3, to visualize apoptotic cells, and phospho-H3, to visualize mitotic cells. As quantified in Supplementary Figure 9f-o and described on page 9-10, there are no significant differences in numbers of binucleated, apoptotic, or mitotic cells between control and Cep55 KO intestine.

We have now also quantified apoptotic and proliferating cells in the brain cortex, lung, heart, liver and kidneys at E13.5. While we observe an increase in apoptotic and proliferating cells in the brain cortex, Cep55 KO does not affect numbers of apoptotic and proliferating cells in the other analysed organs (Fig. 3 and Supplementary Fig. 3, 5 and 6). At birth, except for the brain, the weight of the above tissues is comparable between control and mutant mice (Fig. 2b). Our interpretation of these results is that in the analysed tissues, except for the brain, most cells divide in a Cep55-independent way.

5. When abscission timing is measured, the cumulative plots should be provided in supplementary data (not only means and SDs) for proper comparisons. Duration of abscission in Fig. 3j and 5d would be interesting.

Cumulative plots have now been provided for Fig. 5j (previously Fig. 3j) in Supplementary Fig. 11k; and for Fig. 7d (previously Fig. 5d) in Supplementary Fig. 14d, as suggested. Please note that the abscission time for siChmp2A in WT fibroblasts in Supplementary Fig. 14d (previously Supplementary Fig. 5c) has been corrected due to a previous typo in the calculation formula. This has not affected our conclusions.

6. In video 3, the presented cell does not fail to complete abscission, as stated. Instead, the cell quickly becomes binucleated and later tries to separate the two nuclei by cytofission. This has been described in PMID 23878225. There is thus no cytokinetic bridge.

We thank the reviewer for this point and have changed the text in the legend of Fig. 5g (previously Fig. 3g) and added the cited reference to the text (ref. 41).

7. Videos of GFP-CHMP4b in EpH4 cells and in wt fibroblasts should be provided. GFP-CHMP4b might be present at the midbody in all cells, but only transiently and before abscission. Thus, results on fixed cells might give the false impression that CHMP4B is usually not recruited in wt fibroblasts. This could also depend on culture conditions (see above see PMID 29507669).

To assess the possibility that we may have missed a transient recruitment of CHMP4B at the midbody, we have performed live cell imaging of both WT and Cep55 KO fibroblasts expressing GFP-Chmp4B (Fig. 6m,n and Supplementary videos 7-10) and confirmed the results obtained in fixed samples. We obtained similar results when fibroblasts were grown on poly-l-lysine. We were unfortunately unable to select viable EpH4 clones that expressed suitable levels of GFP-Chmp4B for live cell imaging. Altogether, our results (described on p.13) support the finding that the majority of wt fibroblasts divide without Cep55, and in the absence of Cep55, fibroblasts complete cell division without recruiting Chmp4B at the midbody.

8. Discussion: KIF14 mutations also lead to microcephaly and could be discussed as well (e.g. PMID 28892560 and 30388224). There are also frequent kidney defects and ciliogenesis defects in KIF14 or Citron Kinase mutants. Is it the case in Cep55 mice? If not analyzed, the authors should be cautious not to conclude that there is no defect in kidneys.

We thank the reviewer for raising this point. In the expanded discussion section of the revised manuscript, we have now cited the suggested papers with mention of KIF14 (refs 47 to 54).

Regarding possible kidney defects in Cep55 KO animals, the first pathology report did not highlight any defect in the kidneys of the rare animals that survive postnatally. However, a second report from a different team of pathologists has found subtle defects in the kidney, that are described as “kidney dysmaturity”. We have provided new histological images of the kidneys in Supplementary Figure 2 and describe these defects on page 6.

9. Typo p.8 line 3: Fig. 5e,f

We apologise for this typo. Original Fig. 5e,f is now Fig 7e,f in the revised manuscript, cited on page 14.

Reviewer #2 (Remarks to the Author):

The authors report the surprising result that proliferation and development of most murine tissues occurs reasonably normally in the absence of the ESCRT adaptor, Cep55. Neuronal progenitor cells are the one clear exception, which provides a nice demonstration that the ESCRT pathway is important for abscission in some contexts. These observations are extended to primary cells in culture, where primary fibroblast abscission is convincingly shown to occur in the absence of Cep55, and upon efficient depletion of several key ESCRT-III factors, whereas neuronal progenitors (and murine cell lines) are shown to require Cep55 and the ESCRT pathway for efficient abscission.

Evaluation

This is a high-quality study that reports a surprising result and adds significantly to the growing body of evidence that ESCRT-mediated abscission is less important in vivo than in transformed cell lines in culture. This basic result is important, convincing and should be of general interest to the cell biology community. The experiments also appear to be well performed, well controlled and quite thorough. I therefore do not have suggestions for additional required experiments. My only comment of substance is that I think the authors should be a bit more cautious in entirely dismissing abscission roles for the ESCRT pathway in different contexts. The reported experiments makes it clear that cells in most living tissues can divide without ESCRT recruitment, which is an important advance, but I think it would be surprising if Cep55/ESCRT do not make contributions in any cell types beyond neural progenitors. For example, Tex14, which competitively inhibits the ability of Cep55 to recruit TSG101 and ALIX, is required for stable intercellular bridge formation during germ cell formation (<https://www.ncbi.nlm.nih.gov/pubmed/19020301> and references therein). Moreover, it looks to me as though there may be room for small effects even in fibroblasts because the cultured primary Cep55+/+ fibroblasts appear to replicate slightly faster than their Cep55-/- counterparts (Supplemental Figure 3c), although this effect may or may not be significant because the Cep55+/- fibroblasts are also slower in this experiment. Similarly, 4OHT treatment appears to increase the number of binucleated Cep55 F/F MEFs slightly (Supplemental Figure 3j). None of this is meant to suggest that the results being reported are not significant or surprising, just to suggest that the Discussion (which is generally well written) might be modified slightly to accommodate these possibilities.

We thank the reviewer for the positive evaluation. We agree with the reviewer that germ cells are a very interesting case. Indeed, ALIX and Chmp4B (also known as Shrub) are also required for abscission in the female germ line in *Drosophila* (Refs 20, 21). We have cited the *Drosophila* papers in the introduction.

The reviewer is also correct that 4OHT treatment increases the number of binucleated Cep55 F/F MEFs slightly (Supplementary Fig. 11j; previously Supplemental Figure 3j) and this is now properly commented on page 10-11. It is therefore possible that some other cells both in vivo and in vitro might divide in a Cep55 dependent way. We have now mentioned this possibility in the discussion (page 14).

Reviewer #3 (Remarks to the Author):

The manuscript submitted by Dr. Tedeschi and co-Authors addresses the in vivo function of Cep55 in mouse development. In contrast with a supposedly generalized function of Cep55 in cytokinesis completion, they show that this protein is mostly required for brain development. Cep55 loss results in microcephaly and reduced weight at birth, but embryo morphogenesis occurs quite normally. Neural progenitors show abnormal cytokinesis and elevated levels of apoptosis. In contrast, embryo fibroblasts divide normally. Authors also show that, besides Cep55, even other ESCRTIII components are not required for cytokinesis of primary fibroblasts. The results are consistent with the phenotype produced in humans by CEP55 loss of function mutations.

Overall, this is a very interesting paper, supporting the view that cytokinesis of neural progenitors is regulated differently from other cell types, as well as that many proteins thought to be generally important for cytokinesis are, in fact, dispensable in most cases.

We thank the reviewer for his positive evaluation.

I have no major issues to raise but, in my opinion, addressing the following points would make the paper stronger:

1. The histological panels of Fig. 1 (e and h) could be significantly improved. The ko section appears significantly more lateral than control, giving the impression of a telencephalic reduction much higher than quantified. For this reason, even the higher power fields are not really comparable.

We thank the reviewer for highlighting this issue. We have re-examined the HE stained sections from P0 brains and chosen more comparable ones which are shown now in Fig. 2d and g. We have updated the quantifications from the original Figure 1 accordingly in Fig. 2 e, f and h.

2. Since a significant number of mice seems to survive to P21, it would be nice to see para-sagittal brain sections at P14. This would allow a better description of which brain regions are most affected by Cep55 loss.

We apologise that this was unclear in the original manuscript. The majority of Cep55 KO pups were cannibalized by their parents soon after birth. Of the seven Cep55 KO animals that survived beyond P0, two had to be sacrificed at P1, two at P9 and three at P14 owing to motor coordination problems and weight loss. We have now clarified this in the legend of Table 1. The rarity of these survivors meant that we were unfortunately unable to obtain additional Cep55 KO animals at P14. We have improved the description of the fate of Cep55 KO animals to clarify this revision point (page 5).

3. End of page 4, beginning of 5: I think the authors mean cortical plate, not preplate. It would be nice showing at least an example of doublets within a high magnification

field of the cortex.

We thank the reviewer for this correction. We have corrected the text to “cortical plate” (page 8) and we now show examples of NeuN positive doublets in Fig. 4g.

4. Since mice are smaller at birth, some defects must be present also in other cell types. I would strongly suggest a more careful analysis of embryo sections stained for C3A since, to my eye, it seems that some apoptotic cells are present, in different districts. The organization of kidney does not look exactly identical to control. This is quite relevant, since kidney alterations are observed in patients. Flow cytometry analysis of other tissues could also be helpful, to reveal eventual cell cycle abnormalities. In any case, Authors should at least discuss the possible reasons that make ko animals smaller at birth.

We thank the reviewer for highlighting this interesting issue. Following our investigations in response to this comment, we agree that kidney alterations and apoptosis in tissues other than the brain likely contribute to the reduced body weight in Cep55 newborn mice.

We found that the weights of kidneys, lungs, heart and liver were similar in control and Cep55 KO newborn mice as quantified in Fig. 2b. To exclude variations in the amount of milk in the stomach at P0, we also weighed E18.5 embryos. As shown in Fig. 2b and explained on page 9, there were still slight but significant differences in the body weights of control and Cep55 KO embryos even when we removed the brain. In addition to the C3A positive cells previously found in the cortex, C3A staining of whole embryos at E13.5 showed apoptotic cells in the spinal ganglia (Supplementary Fig. 3c). Further analysis of C3A-stained sections by two veterinary pathologists showed “abundant and widespread labelling of cells in the brain (forebrain and midbrain), spinal cord, trigeminal and dorsal root ganglia and more frequent positive cells in the tongue, and around the primitive larynx” in Cep55^{-/-} embryos (Supplementary Note and Supplementary Figure 5). The pathologists also report “kidney dysmaturity” and “small numbers of caspase positive cells” in the kidneys of Cep55 KO animals (Supplementary Fig 2 and 5). These results are now described on pages 6 and 9.

5. It would be nice showing in vivo protein loss, in addition to decreased RNA in the KO. In theory, this could be easily done by WB of developing brain. Added to the supplementary Fig. 1d, it would also increase the confidence that Cep55 is really undetectable in fibroblasts.

We thank the reviewer for this suggestion. We have repeated the Cep55 blot with a newly identified antibody (see Methods) on fibroblast and cortex lysates from different stages of development (Figure 1e, Figure 2c and Supplementary Figure 3b). A clear band around 50 kDa is now detected in control but not in Cep55 KO samples.

6. Authors are very careful in not stating that apoptosis is a consequence of cytokinesis failure, although they show that it frequently occurs in doublets. I suspect cell death could be at least partially not dependent from cytokinesis failure. However, since they have time lapse recordings of knockout divisions, it would be very nice to

assess whether cell death occurs in non dividing cells, before anaphase or only after cleavage furrow ingression. In any case, I think they should at least comment in the discussion about the causal relationships between cytokinesis failure and apoptosis.

This is a very interesting point. We have reanalysed the long-term movies for cell death in non-dividing neural progenitors (Supplementary Fig. 10d and e). We indeed observe that a significant number of Cep55 KO NPCs die in interphase. It is known that in zebrafish and certain cancer cell types Cep55 is involved in regulating the stability/activity of the AKT prosurvival kinase (Refs 55 and 56). This observation and the relationships between cytokinesis failure and apoptosis are discussed on page 15.

7. Apoptosis is most abundant in the intermediate zone, raising the possibility that it may occur in neurons. Authors may want to consider this in their model.

This is an interesting possibility, which we mention in the revised version (p.7).

8. The title of Fig. 4 is not fully justified. In wild type fibroblasts, ESCRT is not absent, but can be detected in approximately 25% of midbodies. The signal is specific, since there is a clear difference with the knockout. For this reason, the title should be smoothed.

Ferdinando Di Cunto

We agree and have replaced the title of Fig 6 (previous Fig. 4) with “ESCRTs are recruited at the MB of neural progenitors but are absent in Cep55 knockout fibroblasts”.

Reviewer #4 (Remarks to the Author):

Tedeschi et al show here that Cep55 knockout neonates present with microcephaly but without significant impairment of other tissues. Lack of Cep55 leads to apoptotic and binucleated cortical cells, due to a requirement for Cep55 in neural progenitor cell abscission. Additionally they show that Chmp2b and Chmp4b are recruited to the midbody of neural progenitors, and that knockdown of multiple ESCRT-III components leads to abscission failure in neural progenitors but not primary fibroblasts.

While the data are interesting, in showing a specific requirement for Cep55 in cortical development through abscission of neural progenitor cells, there are several major and minor issues with the manuscript as listed below that I think would need to be carefully addressed before the manuscript is suitable for publication. Additionally, the novelty of this paper is somewhat compromised by the 2017 publication from Laporte et al, showing that Alix is required during development for normal growth of the mouse brain.

Major concerns:

Specifically the lack of proof of knockdown at the protein level throughout the paper.

We have now tested several new antibodies and included additional western blots showing absence of Cep55 in the knockout samples (please see new Figures 1e and 2c, and Supplementary Figures 1d, 1f, 3b, 9c, 10c, and 11h).

Knockdown of ESCRT components Chmp2A, Chmp2B, and Chmp4B is provided at the mRNA level, and is over 90% in all cases (Supplementary Fig. 14a; original Supplementary Fig 5a). We have tested several additional antibodies for ESCRT components but none give a specific signal in TTFs by Western blot. For example, levels of Chmp4C in TTFs appear to be extremely low in comparison to those in Eph4 cells (Supplementary Fig. 15d).

Throughout the manuscript all stats have been performed using unpaired t-tests. This is not appropriate when more than 2 sets of data are present on any graph, as occurs in numerous figures. Appropriate statistical tests (one or two way ANOVAs) need to be performed for the appropriate graphs.

We apologise for this oversight, and have now performed ANOVAs for the appropriate graphs in which more than two sets of data are compared, as noted in the figure legends.

The discussion needs substantial expansion, and reference therein should be made to the above mentioned Laporte paper.

In response to this comment, and that of Reviewer 1, we have now expanded the discussion. The 2017 publication from Laporte et al was cited in the original manuscript (original Ref 13) and is also discussed in the revised version (ref 25). It is indeed very interesting that ALIX and Cep55 are both required for the development of the mouse brain and we have now highlighted this point. The brain defects in the study by Laporte et al are reported to be due to ALIX requirement for endocytosis in neural progenitors, rather than cytokinesis failure. We postulate that in the absence of ALIX, Cep55-Tsg101 might be able to recruit ESCRT-III at the midbody of neural progenitors in order to complete cell division whereas in the absence of Cep55, ESCRT-III is not recruited.

Major Points:

1. The validation of Cep55 knockdown in the mouse model is not complete. Although knockdown is shown at the mRNA level in Supplementary Figure 1 b, Supplementary Figure 1 d does not prove absence of Cep55 protein in fibroblasts. This blot has a very high background making lanes 3-7 extremely unclear, but as far I can see no Cep55 is visible in +/+ or +/- lanes.

We agree that the blot in Supplementary Figure 1d was unclear. We have now repeated the Cep55 blot with a new antibody (see Methods). A clear band around 50 kDa is now detected in control fibroblasts but not in Cep55 KO fibroblasts and shown in Fig. 1e and Supplementary Fig. 1d.

In addition, we have now provided a scheme showing conservation of the Cep55 ESCRT- and ALIX-binding region (EABR) and midbody localization domains in

human and mouse (Supplementary Fig. 1a). Both these domains are encoded by exons that are deleted, or located downstream of stop codons, in the Cep55 null alleles tm1a and tm1d (Fig. 1a). Genotyping PCRs in Fig. 1b and Supplementary Fig. 1b confirm these alleles, and quantitative PCRs to validate knockdown of Cep55 mRNA using several primer pairs are shown in Supplementary Fig. 1c.

Furthermore, knockdown should be shown at the protein level in multiple tissues, as several different tissues are examined during this study (Supplementary Figure 2). This is of course extremely pertinent for brain tissues.

New Fig. 2c and Supplementary Fig. 3b show Cep55 western blots from microdissected brain cortex lysates at different developmental stages. At all stages Cep55 is undetectable in the $-/-$ genotype.

Micro-dissected lungs, liver, gut, heart and kidneys from E19.5 animals also show Cep55 depletion in the $-/-$ genotype (Supplementary Fig. 1f). Supplementary Fig. 9c shows Cep55 depletion from proliferative crypts of intestinal epithelium in Cep55^{ΔG/-} Villin-Cre animals.

Additionally, proof of knockdown at the protein level would be more appropriately shown in Figure 1, with the uncropped blots shown in full in the supplemental to prove no protein truncation is occurring.

We now present the blot showing loss of Cep55 protein in the knockout in Fig. 1e, as suggested. The uncropped blots are shown in the Source Data file, as requested by the journal.

2. “Together, these data indicate that Cep55 deletion promotes microcephaly as result of cell death by apoptosis.” From the representative image (Fig 2d) it looks like the cleaved caspase 3 staining seems to be restricted largely to the upper SVZ and IZ. This should be noted in the text. Has this been quantified?

We thank the reviewer for raising this interesting point. In response to this comment, we have now defined the borders between the cortical layers more precisely in the representative image in Fig. 3e (previously Fig. 2d; see Methods) and quantified the distribution of C3A+ cells through the different layers at E13.5 and E16.5 in Fig. 3 h. We indeed find that the SVZ and the IZ are the most affected layers, as described on page 7.

3. Patients with Cep55 mutations present with brain and kidney defects. Supplementary Figure 2 d claims that HE staining of kidney shows no defect, however the tissue distribution looks different between genotypes in this representative image. Please amend or comment on this phenomenon. As per major point 1, Cep55 depletion in these investigated tissues has not been verified.

Cep55 depletion in the investigated tissues has now been verified by western blotting, as described in our response to major point 1, above.

The reviewer is right about the organization of the kidney. In response to this comment and those from reviewers 1 and 3, we requested a second opinion on the histology of the kidney and the other main organs in the rare P14 mutant mice from two veterinary pathologists, whose report is now included as a Supplementary Note. The pathologists report “kidney dysmaturity” in Cep55 KO animals. This is now documented in Supplementary Fig. 2 and discussed on page 6 in relation to the human defects.

4. Supplementary Figure 3 a, b: again, protein blots to quantify knockdown are needed here, as this is not a full knockdown at the mRNA level (Supplementary figure 3b- appropriate stats need to be performed and included).

We have now quantified Cep55 protein knockdown of over 99% on western blots using cells from the experiment in original Supplementary Fig 3a, b (now Supplementary Fig. 10a, b, c). We have also performed one-way ANOVA followed by Tukey's multiple comparisons test for the qPCR results in Supplementary Fig. 10b (original Supplementary Fig. 3b) and the knockdown is significant.

5. Figure 4 d shows bright cytosolic punctate Cep55 staining in Cep^{-/-} fibroblasts. What is this?

We believe that the punctate cytosolic staining in Fig. 4d (now Fig 6d) is non-specific signal and have noted this in the legend. The fibroblasts (tm1d) used here have now been fully validated by WB and qPCR in Supplementary Fig.1c-d, so we are confident that the signal is not from Cep55. We have additionally provided Cep55 IF quantification along the intercellular bridge in Fig. 6e.

6. Fig 3 c does not show a significant increase in pyknotic cells in treated cells compared to vehicle control ($p = 0.298$), although the text states there is a significant increase. Is this a typo on the graph?

We apologise for this typo, which should have read $p=0.0298$. In response to the concern above, we have now performed a one-way ANOVA followed by Tukey's multiple comparisons test and the p value is noted in the legend of Fig. 5c (original Fig. 3c).

7. Chmp4C^{-/-} fibroblasts. Methods state these KO mice were generated for this study. However, no validation of the model is presented here. This model needs to be validated and appropriate data showing Chmp4c specific knockdown needs to be presented.

The Chmp4C^{tm1a} allele has previously been validated by the International Mouse Phenotyping Consortium (ref 64). We generated mice using sperm provided by EUCOMM carrying this validated Chmp4C^{tm1a} allele. We now describe the allele and our validation of Chmp4C KO in Supplementary Figure 15. qPCR results for Chmp4C were shown in the original manuscript (Supplementary Figure 5a) and are now in Supplementary Fig. 15 with additional primers. Using primers spanning

Chmp4C exons that translate for helices a2-a3, we could not detect any Chmp4C mRNA in knockout fibroblasts (Supplementary Fig. 15e).

8. Fig 5 d and f.

In Fig 5 d it looks like there may be a significant difference between Chmp4c^{+/+} + Ctr siRNA and Chmp4c^{-/-} + Chmp4b siRNA. Is there? All significant differences should be stated.

There is no significant difference in the rates of abscission failure (see below) between Chmp4c^{+/+} +siCtr and Chmp4c^{-/-} +siChmp4b conditions (Fig. 7d, original Fig. 5d). However, there is a significant reduction in the occurrence of furrow regression before intercellular bridge formation (see below). *P* values for both these comparisons, calculated using two-way ANOVA followed by Dunnett's multiple comparisons test, are now given in Fig. 7d.

Additionally, I'm confused by the inclusion of 2 different criteria for abscission failure in Fig 5 d, and only 1 abscission failure for 5 f. It looks like if the binucleated data in 5 d were pooled similarly to 5 f they might be significant. Please comment.

Thank you for the opportunity to clarify this point (page 11). In the "Binucleated/no furrow" category (renamed "Binucleated/Furrow regression" to better describe the phenotype), cells become binucleated as a consequence of incomplete ingression followed by regression of the cleavage furrow, before formation of the intercellular bridge. This only occurs in TTFs (Fig. 7d, original Fig. 5d), not NPCs (Fig. 7f, original Fig. 5f) and we do not consider this to be abscission failure. The "Binucleated/Abscission failure" category only quantifies cells that fail abscission after complete ingression of the cleavage furrow and formation of the intercellular bridge. Since these categories are mutually exclusive, we do not pool them; only the latter category is observed in NPCs. For consistency, we have also included additional images in Supplementary Figure 14c distinguishing "No furrow" (e.g. following MKLP1 knockdown) from "Furrow regression" and "Abscission failure" categories used in the quantification of the fates of dividing fibroblasts following ESCRT knockdown (Supplementary Figure 14b).

Minor Points:

1. Pg 4: "Cep55 was highly expressed in the developing central nervous system (Supplementary Fig. 1e, f), consistent with an essential role in brain development." Only P0 is shown in this figure supporting Cep55 expression in neonates, so this sentence needs to be rephrased or normal levels of Cep55 expression investigated at different timepoints of CNS development.

We thank the reviewer for raising this point. We have now performed Cep55 WB on extracts from micro-dissected cortices from different developmental stages (Fig. 2c and Supplementary Fig. 3b). The results, described on page 6-7, are consistent with an essential role of Cep55 during neurogenesis, particularly at early stages.

2. Figure 2 a and d: Although it is apparent that the layers are thinner, it appears that the correct layering of the cortex is intact. This should be noted in the text.

We agree with the referee and have noted this point in the text (page 6, “the typical layered structure of the cortex was maintained”).

3. “Numbers of Ki67-positive nuclei scored by immunohistochemistry were similar in control and Cep55^{-/-} cortices (control 17.5±2.1% versus Cep55^{-/-} 21±1.7% at E16.5; n = 3470 and 2237 cells respectively, from 3 mice per genotype), suggesting that differences in proliferation rates do not explain the loss of cells in Cep55^{-/-} mice.” This should say percentage instead of numbers; the Cep55^{-/-} brains already have fewer cells (as evidenced by the number of cells per genotype). This should be linked to Fig 2g, which is omitted in the text. Why are these graphs and the stats not shown? The y axes in Fig 2g are not visible, the resolution needs to be improved.

We have now included the Ki67 immunostaining and quantifications for E13.5 and E16.5 in Supplementary Fig. 3d and e. We have corrected “numbers” to “percentage” in the description of these results on page 7 and have redrawn the axes of Fig 2g (now Fig. 4b) to improve the resolution. The reviewer is correct that the percentage of Ki67-positive cells from the FACS at E16.5 in Fig. 4c (originally Fig. 2g), is consistent with the immunostaining, and we have now referred to this in the text (page 8, “The percentage of Ki67-positive cells was not different between control and Cep55 knockout samples (Fig. 4c), consistent with the immunohistochemistry data (E16.5, Supplementary Fig. 3d).

4. “In newborn Cep55^{-/-} mice, 32.2±8.9% of 5 neurons in the cortical preplate layer (mean ±sd; n = 284 cells from 3 mice) also appeared as doublets or were abnormally large compared with 4.7± 2.1% of neurons in control cortices (n = 325 cells), consistent with binucleation and with the known function of Cep55 in abscission.” Where is this data? Or data not shown?

These data were originally given in the text only, but we have now included them together with representative images in Fig. 4g, as requested also by reviewer 3.

For completeness, we have also now included a graph in Supplementary Fig. 11b quantifying the percentage of control and Cep55^{-/-} TTFs at the abscission stage in culture, data which was previously given in the text only.

5. Fig 2J Distance (inches) is not appropriate when the scale bar on the figure from which these inserts are lifted is 2 μm.

We apologise for this oversight and have relabelled the figure (now Fig. 4f) accordingly as “Distance (μm).

6. Pg 5. I’m confused by the change to a conditional knockout at this point. Presumably this is because established full knockout does not produce culturable neural progenitors. If so, please state. Additionally the changes between conditional knockout and Cep^{-/-} fibroblasts from the mouse are confusing here, and could do with clarifying in the text.

The reviewer is correct that we use the conditional allele to study the neural progenitors in vitro because neural progenitors from the full knockout do not survive. We have now explained this on page 10 of the revised manuscript (relating to Supplementary Fig. 10a-c). We have also explained later on page 10 (relating to Supplementary Fig. 11e-j) that we used Cep55 conditional KO fibroblasts because we wanted to test acute deletion of Cep55.

7. "Supplementary Figure 3: Cep55 is dispensable for proliferation and division of primary fibroblasts." This title needs to be amended as it includes data from conditional knockout NPCs as well as primary fibroblasts. Please clarify.

To clarify this, we have now separated the data from conditional knockout NPCs (now Supplementary Fig. 10) and primary fibroblasts (now Supplementary Fig. 11) and titled the figures accordingly.

8. Pg 7 "and that the majority of wild-type fibroblasts divide without Cep55 and without detectable ESCRT-III." However as has been investigated in the next section, ESCRT-III redundancy has not been investigated here, and presence of Chmp2a or Chmp4c cannot be excluded.

The reviewer is correct and we have amended the sentence to read "and that the majority of wild-type fibroblasts divide without Cep55" (page 13).

9. Fig 5 g The representative images for Pax6 staining look quite different between Cep^{+/+} and Cep^{-/-}, although the quantification does not support this. Please choose different representative images.

We apologise that this analysis was not clear. We have now replaced the previous Pax6 staining images (original Fig. 5g) with alternative images from the same sections (new Fig. 3i). Both the original and new images show a reduced thickness of the Pax6-positive cell layer of approximately 50% in the Cep55^{-/-} cortex, as we have now quantified in Fig. 3j, left-hand graph. However, the percentage of Pax6-positive nuclei (as a proportion of total nuclei within the section) is the same (Fig. 3j, right-hand graph, original Fig. 5g).

10. Fig 5 i For the Tbr2 staining, where the number of stained cells are significantly different at E13.5, it looks like there are more positively stained cells in the uppermost layer. Is this real?

We have now replaced the previous Tbr2 staining images (original Fig. 5i) with alternative images from the same sections (new Fig. 3k) and quantified the percentage of Tbr2-positive cells specifically in the SVZ. Indeed, as the reviewer observed, there are more positively stained cells in the uppermost layer (SVZ) in control versus Cep55^{-/-} cortex at E13.5 (Fig. 3l, right-hand graph).

REVIEWERS' COMMENTS:

Reviewer #1 (Remarks to the Author):

The authors have successfully addressed the points I raised. I am happy to recommend publication.

Reviewer #2 (Remarks to the Author):

In my opinion, the authors have done an unusually thorough job in addressing the issues raised by the reviewers, and I believe that the manuscript, in its current form, will be a valuable addition to the cytokinesis literature.

Reviewer #4 (Remarks to the Author):

Having read the revision of the manuscript, I am happy that the authors have very thoroughly responded to each of my initial concerns. I have no further issues to raise, and recommend the study for publication.

We are pleased the reviewers are happy with our revised manuscript and recommend it for publication. We would like to thank the reviewers for their thorough and constructive feedback during the revision process.

REVIEWERS' COMMENTS:

Reviewer #1 (Remarks to the Author):

The authors have successfully addressed the points I raised. I am happy to recommend publication.

Reviewer #2 (Remarks to the Author):

In my opinion, the authors have done an unusually thorough job in addressing the issues raised by the reviewers, and I believe that the manuscript, in its current form, will be a valuable addition to the cytokinesis literature.

Reviewer #4 (Remarks to the Author):

Having read the revision of the manuscript, I am happy that the authors have very thoroughly responded to each of my initial concerns. I have no further issues to raise, and recommend the study for publication.